# ROBUST AND CONTROLLABLE OBJECT-CENTRIC LEARNING THROUGH ENERGY-BASED MODELS

**Ruixiang Zhang**[†]    **Tong Che**[°]    **Boris Ivanovic**[°]    **Renhao Wang**[°]
**Marco Pavone**[°‡]    **Yoshua Bengio**[†]    **Liam Paull**[†]

[°]Nvidia Research

[‡]Stanford University

[†]Mila, Université de Montréal

{ruixiang.zhang, tong.che}@umontreal.ca

## ABSTRACT

Humans are remarkably good at understanding and reasoning about complex visual scenes. The capability to decompose low-level observations into discrete objects allows us to build a grounded abstract representation and identify the compositional structure of the world. Accordingly, it is a crucial step for machine learning models to be capable of inferring objects and their properties from visual scenes without explicit supervision. However, existing works on object-centric representation learning either rely on tailor-made neural network modules or strong probabilistic assumptions in the underlying generative and inference processes. In this work, we present EGO, a conceptually simple and general approach to learning object-centric representations through an energy-based model. By forming a permutation-invariant energy function using vanilla attention blocks readily available in Transformers, we can infer object-centric latent variables via gradient-based MCMC methods where permutation equivariance is automatically guaranteed. We show that EGO can be easily integrated into existing architectures and can effectively extract high-quality object-centric representations, leading to better segmentation accuracy and competitive downstream task performance. Further, empirical evaluations show that EGO's learned representations are robust against distribution shift. Finally, we demonstrate the effectiveness of EGO in systematic compositional generalization, by re-composing learned energy functions for novel scene generation and manipulation.

## 1 INTRODUCTION

The ability to recognize objects and infer their properties and relations in a scene is a fundamental capability of human cognition. The central question of how objects are discovered and represented in the brain has been a subject of intense research for decades, and has prompted the field of cognitive science (Spelke, 1990) to ask how we might develop intelligent machine agents to learn to represent objects in the same way humans do, without being explicitly taught what those objects are. Developing artificial agents capable of decomposing complex scenes into discrete objects can be a crucial step for many applications in robotics, vision, reasoning, and planning. Learning such object-centric representations can further help to identify the relational and compositional structure among objects and enables the agent to reason about a novel scene composed of new objects by leveraging knowledge from previously-learned representations of similar objects.

In recent years, many works have been proposed to learn object-centric representations from visual scenes without human supervision. A variety of models, in the form of structured generative models (Greff et al., 2019; Burgess et al., 2019; Engelcke et al., 2020; Lin et al., 2020) or specifically designed neural network modules (Locatello et al., 2020), have been proposed to tackle the problem of visual scene decomposition and generation. On the other hand, recent progress in large language models (Vaswani et al., 2017; Brown et al., 2020) and visual-language models (Radford et al., 2021; Ramesh et al., 2022) shows the huge potential of training expressive neural network models with

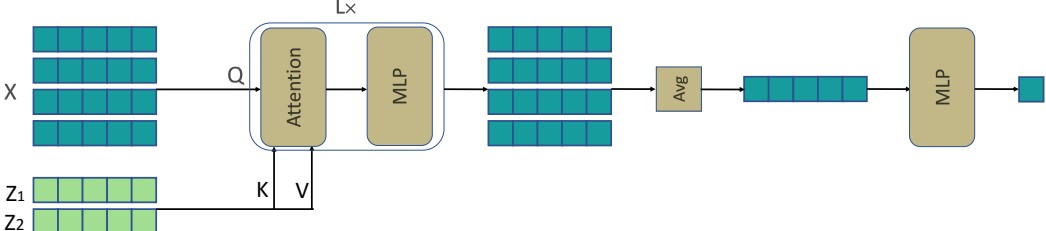

Figure 1: Architecture of EGO-Attention, the variant of EGO used in experiments. $\mathbf{x}$ is the input image and $\mathbf{z}_i$ are the object-centric representations. In each block, EGO attends to the latent variables to refines the hidden scene representation using cross-attention mechanism between $\mathbf{x}$ and $\mathbf{z}_i$, to measure the consistency between image input and latent representation.

minimal hand-designed inductive biases. In a similar spirit, we ask whether we can learn object-centric representations with minimal human assumptions and task-specific architectures.

**Contributions**   In this work, we introduce EGO (**E**ner**G**y-based **O**bject-centric learning), a conceptually simple yet effective approach to learning object-centric representations without the need for specially-tailored neural network architectures or strong (typically parametric) assumptions on data generating process. Based on the Energy-based Model (EBM) framework, we propose to learn an energy function that takes as input a visual scene and a set of object-centric latent variables and outputs a scalar value that measures the consistency between the observation and the latent representation (Section 2). We minimally assume permutation invariance among objects and embed this assumption into the energy function by leveraging the vanilla attention mechanisms from the Transformer (Vaswani et al., 2017) architecture (Section 2.1). In essence, our method makes models act as segmentation annotators, aiming to iteratively improve their annotations by minimizing our energy function. We use gradient-based Markov chain Monte Carlo (MCMC) sampling to efficiently sample latent variables from the EBM distribution, which automatically yields a permutation-equivariant update rule for the latent variables (Section 2.2). This stochastic inference procedure also addresses the inherent uncertainty in learning object-centric representations; models can learn to represent scenes containing multiple objects and potential occlusions in a probabilistic and multi-modal manner. We demonstrate the effectiveness of our approach on a variety of unsupervised object discovery tasks and show both qualitatively and qualitatively that our model can learn to decompose complex scenes into highly accurate and interpretable objects, outperforming state-of-the-art methods on segmentation performance (Section 4.1). We also show that we can reuse the learned energy functions for controllable scene generation and manipulation, which enables systematic compositional generalization to novel scenes (Section 4.2). Finally, we demonstrate the robustness of our model to various distribution shifts and hyperparameter settings (Section 4.3).

## 2   ENERGY-BASED OBJECT-CENTRIC REPRESENTATION LEARNING

The goal of object-centric representation learning is to learn a mapping from a visual observation $\mathbf{x} \in \mathbb{R}^{D_\mathbf{x}}$ to a set of vectors $\{\mathbf{z}_k\}$, where each vector $\mathbf{z}_k \in \mathbb{R}^{D_\mathbf{z}}$ describes an individual object (or background) in $\mathbf{x}$. In this work, we make use of an EBM $E(\mathbf{x}, \mathbf{z}; \boldsymbol{\theta})$, parameterized by $\boldsymbol{\theta}$, to learn a joint energy function which assigns low energy to regions where the visual observation $\mathbf{x}$ and the latent object descriptors $\mathbf{z}$ are consistent, where $\mathbf{z} = \{\mathbf{z}_k\}_{k=1}^K$ are a set of $K$ object-centric latent variables. To implement the mapping from a visual scene to its constituting objects, we can sample from the posterior distribution $\mathbf{z} \sim p(\mathbf{z}|\mathbf{x}; \boldsymbol{\theta}) \propto e^{-E(\mathbf{x}, \mathbf{z}; \boldsymbol{\theta})}$ by using any efficient MCMC sampling algorithm, such as the stochastic gradient Langevin dynamics method (Parisi, 1981; Welling & Teh, 2011) and Hamiltonian Monte Carlo (Duane et al., 1987; Neal et al., 2011). Accordingly, the EBM $E$ can be used as a generic module for object-centric representation learning, offering great flexibility in which neural network architectures can be used and the functional form of the energy function.

### 2.1   PERMUTATION INVARIANT ENERGY FUNCTION

One fundamental inductive bias in object-centric representation learning is encoding the permutation invariance of a set of objects into model learning. In this section, we introduce two formulations

of the energy function $E(\mathbf{x}, \mathbf{z}; \boldsymbol{\theta})$ that are permutation invariant with respect to the order of the object-centric latent variables $\{\mathbf{z}_k\}$.

**EGO by composing individual energy functions**    First, we consider a simple formulation of the energy function $E(\mathbf{x}, \mathbf{z}; \boldsymbol{\theta})$ that is decomposed into a set of individual energy functions $E(\mathbf{x}, \mathbf{z}_k; \boldsymbol{\theta}_k)$, followed by a permutation invariant aggregation function $\phi$ as $E(\mathbf{x}, \mathbf{z}) = \phi\left(\{E(\mathbf{x}, \mathbf{z}_k)\}\right)$.

The individual energy function $E(\mathbf{x}, \mathbf{z}_k; \boldsymbol{\theta}_k) : \mathbb{R}^{D_{\mathbf{x}}} \times \mathbb{R}^{D_{\mathbf{z}}} \mapsto \mathbb{R}$ can be any function that takes the observation $\mathbf{x}$ and a single latent variable $\mathbf{z}_k$ as input, and outputs a scalar energy value which quantifies the belief that an object with representation $\mathbf{z}_k$ is present in the visual scene $\mathbf{x}$. We share the parameters $\boldsymbol{\theta}_k$ across all the individual energy functions $\theta_k = \theta \ \forall k$, such that it can generalize to an arbitrary number of objects without breaking symmetry.

The aggregation function $\phi$ is a permutation-invariant function with respect to the set $\{E(\mathbf{x}, \mathbf{z}_k)\}$. We can use any function $\phi$ that is invariant to the order of inputs, such as the sum (Du et al., 2020; 2021), minimum (Parascandolo et al., 2018), or parameterized transformations (Zaheer et al., 2017). Throughout this work, we use the sum as the aggregation function, which is effective in encouraging the model to learn to decompose the input into discrete objects and local variations, as explored in (Du et al., 2021; Zhang et al., 2020). We call the resulting EBM formulation `EGO-Sum`, given by

$$E(\mathbf{x}, \mathbf{z}; \boldsymbol{\theta}) = \sum_{k=1}^{K} E(\mathbf{x}, \mathbf{z}_k; \boldsymbol{\theta}) \tag{1}$$

**EGO from permutation equivariant/invariant transformations**    We introduce another formulation of the energy function $E(\mathbf{x}, \mathbf{z}; \boldsymbol{\theta})$ that composes multiple permutation equivariant/invariant transformations on top of the set $\{\mathbf{z}_k\}$ and $\mathbf{x}$. In particular, we use vanilla attention blocks (Vaswani et al., 2017), such as cross-attention and self-attention, to build up these differentiable mappings.

We first encode the data input $\mathbf{x} \in \mathbb{R}^{D_{\mathbf{x}}}$ into a higher-level representation $\mathbf{h} \in \mathbb{R}^{N_{\mathbf{h}} \times D_{\mathbf{h}}}$, where $N_{\mathbf{h}} = \text{Width}_{\mathbf{h}} \times \text{Height}_{\mathbf{h}}$ is the number of vectors in the 2D feature map $\mathbf{h}$ when using a convolutional neural network (CNN) as the backbone encoder.

We then use a stack of $L$ standard transformer blocks with a cross-attention layer to fuse the information in $\mathbf{h}$ and the object-centric latent variables $\mathbf{z} = \{\mathbf{z}_k\}_{k=1}^{K}$. Each block consists of a multi-head cross-attention layer, followed by a position-wise, fully connected feed-forward network. In each cross-attention layer, a linear transformation is applied to the image feature map $\mathbf{h}$ to produce queries over the set of latent variables. Cross-attention weights are then computed between the queries and linearly-projected keys and values from the set of latent variables $\{\mathbf{z}_k\}$. The stacked transformer blocks allow the model to sufficiently capture information from $\{\mathbf{z}_k\}$ by attending to the most relevant subset of latent variables at each image feature location, such that the model can learn to tell whether the set of latent variables fully explains each object.

We then use an average pooling layer to aggregate the final output of the transformer blocks $\mathbf{h}_L \in \mathbb{R}^{N_{\mathbf{h}} \times D_{\mathbf{h}}}$, into a single vector, which is then passed through a fully-connected layer to produce the scalar energy term.

We provide an overview of our architecture in Figure 1, and call the resulting EBM formulation `EGO-Attention`, given by:

$$\begin{align}
\mathbf{h}_0 &= \text{Encoder}(\mathbf{x}) \tag{2}\\
\mathbf{h}'_\ell &= \text{CrossAttention}(\text{LayerNorm}(\mathbf{h}_{\ell-1}), \text{LayerNorm}(\mathbf{z})) + \mathbf{h}_{\ell-1} \qquad \ell = 1 \dots L \tag{3}\\
\mathbf{h}_\ell &= \text{MLP}(\text{LayerNorm}(\mathbf{h}'_\ell)) + \mathbf{h}'_\ell \qquad \ell = 1 \dots L \tag{4}\\
E &= \text{MLP}(\text{AvgPool}(\mathbf{h}_L)) \tag{5}
\end{align}$$

## 2.2    LEARNING AND INFERENCE

For tasks requiring a geometric understanding of a visual scene and reasoning over entities, our model can be used as a plug-and-play module to be integrated into existing architectures for encoding structured object-centric representations. Owing to the great flexibility of `EGO`'s energy function

---

**Algorithm 1:** Training procedure of `EGO` for unsupervised object discovery.

---

**Input:** Image data $\mathbf{x} \in \mathbb{R}^{D_\mathbf{x}}$, number of latent variables $K$, number of MCMC iterations $T$, step size $\epsilon$

**Parameters:** `EGO` $E(\mathbf{x}, \mathbf{z}; \boldsymbol{\theta})$, decoder $\mathrm{Decoder}(\mathbf{z}; \boldsymbol{\phi})$

**Output:** Training loss for unsupervised object discovery

  `# Infer object-centric latent variables by Langevin MCMC sampling`

Draw random initialization $\mathbf{z}^0 \sim \mathcal{N}(\mathbf{0}, \mathbf{I})$

**for** $t = 0$ *to* $T - 1$ **do**

    `# Using permutation-invariant energy functions from Eq. 1 or Eq. 5`

    $\boldsymbol{\eta}^t \sim \mathcal{N}(\mathbf{0}, \mathbf{I})$

    $\mathbf{z}^{t+1} = \mathbf{z}^t + \epsilon \nabla_\mathbf{z} E(\mathbf{x}, \mathbf{z}^t; \boldsymbol{\theta}) + \sqrt{2\epsilon} \boldsymbol{\eta}^t$

`# Decode the latent variables to create reconstructions`

$\tilde{\mathbf{x}} = \mathrm{Decoder}(\mathbf{z}^T; \boldsymbol{\phi})$

`# Compute the reconstruction loss`

**return** $\mathcal{L}_{\mathrm{rec}} = \frac{1}{D_\mathbf{x}} \|\tilde{\mathbf{x}} - \mathbf{x}\|^2$

---

and learning objective formulation, we can customize the model to adapt to a wide range of task contexts and learning desiderata.

Among many possible training objective choices (e.g., maximum likelihood training with MCMC sampling or Contrastive Divergence (Hinton, 2002)) to learn the model as a monolithic generative model, akin to the family of existing approaches (Engelcke et al., 2020; Greff et al., 2019; Burgess et al., 2019) for visual scene understanding and generation, we focus on investigating the potential of `EGO` as a generic standalone module for extracting object-centric representations, similar to (Locatello et al., 2020). To this end, we adopt an encoder-decoder architecture, with our `EGO` module serving as the encoder to transform the unstructured observation into structured object representations, which are then decoded by a separate decoder into reconstructions or other task-specific predictions.

**Encoding object-centric representations by MCMC sampling**    To infer the set of object-centric latent variables $\mathbf{z}$ from the input $\mathbf{x}$, we use gradient-based MCMC sampling methods to sample from the posterior distribution $\mathbf{z} \sim p(\mathbf{z}|\mathbf{x}) \propto e^{-E(\mathbf{x}, \mathbf{z}; \boldsymbol{\theta})}$. Specifically, in this work we utilize the Langevin MCMC (Parisi, 1981; Welling & Teh, 2011) method. Starting from a random initialization $\mathbf{z}^0$ drawn from a simple prior distribution, we iteratively update the latent variables by simulating the Langevin diffusion process for $T$ steps, with step size $\epsilon$, as follows:

$$\mathbf{z}^{t+1} = \mathbf{z}^t + \epsilon \nabla_\mathbf{z} E(\mathbf{x}, \mathbf{z}^t; \boldsymbol{\theta}) + \sqrt{2\epsilon} \boldsymbol{\eta}^t, \ t = 0, 1, \ldots, T-1, \ \boldsymbol{\eta}^t \sim \mathcal{N}(\mathbf{0}, \mathbf{I}), \ \mathbf{z}^0 \sim \mathcal{N}(\mathbf{0}, \mathbf{I}) \quad (6)$$

where $\mathbf{z}^t$ denotes the latent variables at the $t$-th iteration. When $\epsilon \to 0$ and $T \to \infty$, the sampling process converges to the true posterior distribution $p(\mathbf{z}|\mathbf{x})$ under some regularity conditions.

Though running MCMC sampling until convergence can be computationally expensive, we are simulating the Langevin dynamics in the latent space $\mathbf{z} \in \mathbb{R}^{K \times D_\mathbf{z}}$ rather than the high-dimensional pixel space, in contrast to previous works (Du et al., 2021; Du & Mordatch, 2019). We also only run Langevin dynamics for a relatively small number of iterations ($T < 10$) and find that it is sufficient to produce good latent variable samples $\mathbf{z}^T$ in our experiments. This allows us to make use of the gradient-based MCMC sampling in a much more efficient manner, even comparable to amortized inference methods (Eslami et al., 2016; Greff et al., 2017; 2019; Burgess et al., 2019).

**Training procedure**    We outline the detailed training procedure in Algorithm 1, taking the unsupervised object discovery task as an example.  Given the encoded structured representation $\mathbf{z}^T \sim p(\mathbf{z}|\mathbf{x}; \boldsymbol{\theta})$ from MCMC sampling, we use a decoder to map the set of latent variables to task-specific predictions. For unsupervised object discovery, we follow prior approaches and use a spatial broadcast decoder (Watters et al., 2019; Greff et al., 2019; Locatello et al., 2020) to decode each latent variable $\mathbf{z}_k$ separately into an alpha mask $\boldsymbol{\alpha}_k \in [0, 1]^{D_\mathbf{x}}$ and input reconstruction $\tilde{\mathbf{x}}_k$, which are then combined to produce the final output $\tilde{\mathbf{x}} = \sum_{k=1}^K \mathrm{Softmax}(\boldsymbol{\alpha})_k \tilde{\mathbf{x}}_k$. The model can be trained end-to-end to minimize the reconstruction loss. For other tasks, we can use corresponding

decoders (Kosiorek et al., 2020; Zhang et al., 2019; Zaheer et al., 2017) to map latent variables to predictions and train the model end-to-end to minimize task-specific losses.

## 3 RELATED WORK

**Object-centric learning**   Object-centric representation learning (Greff et al., 2015; 2016; Eslami et al., 2016; Greff et al., 2017; 2019; Lin et al., 2020; Xie et al., 2021) plays an important role in scene understanding, visual reasoning, and compositional generalization. Many research works like MONet (Burgess et al., 2019), IODINE (Greff et al., 2019), GENESIS (Engelcke et al., 2020; 2021), and SPACE (Lin et al., 2020) propose various probabilistic models to build a spatial mixture model of visual scenes and use variational inference to learn object-centric latent variables. Slot attention (Locatello et al., 2020) proposed a novel inverted attention mechanism to iteratively assign objects to slots, by introducing competition among slots via attention weight normalization. (Kipf et al., 2021) further applied the slot attention module to video data, and (Chang et al., 2022) improved the approach by using fixed points as object representations, yielding better stability. Besides reconstruction-based object-centric learning, various works (Racah & Chandar, 2020; Kipf et al., 2020; Löwe et al., 2020; Pirk et al., 2019; Baldassarre & Azizpour, 2022) tackle the problem from the perspective of contrastive learning and self-supervised learning. Object-centric representations are also explored in other tasks such as visual question answering (Huang et al., 2020; Dang et al., 2021), visual reasoning (Assouel et al., 2022), and neural scene rendering (Guo et al., 2020).

**Set generation**   Set generation (Vinyals et al., 2016; Rezatofighi et al., 2018; Zhai et al., 2020; Rezatofighi et al., 2021) has received much attention in computer vision and natural language processing tasks in the past few years. (Kosiorek et al., 2020) explored using a cardinality-conditioned transformer for set prediction. The DETR (Carion et al., 2020) model also adopted a transformer to predict a set of objects for the object detection task. The Deep Set Prediction Network (DSPN) (Zhang et al., 2019) proposed a set prediction approach by using a gradient optimization inner-loop with a permutation-invariant encoder to iteratively update the representation of the target set prediction by minimizing the distance between the input and output in a latent space induced by the encoder. Though the bi-level optimization framework is similar to ours, the DSPN model uses the deterministic gradient descent procedure to optimize a pre-defined loss function, while our formulation uses stochastic MCMC sampling to optimize a learned energy function. Our formulation brings more flexibility to the optimization process, and the introduced stochasticity also results in more robustness and diversity in both inference and generation (Section 4.3).

**Energy-based models**   Energy-based models (EBMs) (LeCun et al., 2006) have been widely used in learning abstract concepts (Du et al., 2020) and controllable generation (Nie et al., 2021; Che et al., 2020), due to the compositionality nature of the energy function (Andreas, 2019). (Mordatch, 2018) proposed an EBM-based framework for learning abstract concepts from observations, which enables compositional generalization by test-time optimization. (Zhang et al., 2020) explored the approach of using EBM to replace the hand-crafted permutation-invariance loss functions which aim to optimize the prediction output towards observable ground-truth data, it differs from our work in that we focus on learning the latent object-centric representations from the EBM for unsupervised object discovery and controllable scene manipulation. (Du et al., 2021) also proposed to use EBM to learn both local and global factors of variations from image data, where they proposed to train the model by a nested gradient descent optimization in the high-dimensional pixel space, which can be much more computationally expensive compared to our model which runs Langevin dynamics in the lower-dimensional latent space.

## 4 EXPERIMENTS

### 4.1 UNSUPERVISED OBJECT DISCOVERY

We quantitatively and qualitatively evaluate our proposed model on the task of unsupervised object discovery, with the goal of decomposing the visual scene into a set of objects without any human supervision. As we will show, our approach is able to consistently segment images into highly interpretable and meaningful object masks.

Table 1: ARI scores for unsupervised object discovery on CLEVR-6, Multi-dSprites, Tetrominoes datasets.

|  | CLEVR6 | Multi-dSprites | Tetrominoes |
|---|---|---|---|
| Slot MLP (Locatello et al., 2020) | $60.4 \pm 6.6$ | $60.3 \pm 1.8$ | $25.1 \pm 34.3$ |
| MONet (Burgess et al., 2019) | $96.2 \pm 0.6$ | $90.4 \pm 0.8$ | — |
| Slot-Attention (Locatello et al., 2020) | $\mathbf{98.8 \pm 0.3}$ | $91.3 \pm 0.3$ | $\mathbf{99.5 \pm 0.2}$ |
| IODINE (Greff et al., 2019) | $\mathbf{98.8 \pm 0.0}$ | $76.7 \pm 5.6$ | $99.2 \pm 0.4$ |
| EGO-Sum | $96.1 \pm 0.4$ | $82.1 \pm 0.7$ | $99.1 \pm 0.3$ |
| EGO-Attention | $\mathbf{98.9 \pm 0.5}$ | $\mathbf{93.8 \pm 0.5}$ | $\mathbf{99.6 \pm 0.2}$ |

**Datasets**   In line with previous state-of-the-art works on object discovery, we use the following three multi-object datasets (Kabra et al., 2019): CLEVR (Johnson et al., 2017), Multi-dSprites (Matthey et al., 2017), and Tetrominoes (Greff et al., 2019). We also use a variant of the CLEVR dataset which filters out scenes with more than 6 objects, referred to as CLEVR-6. Same as IODINE (Greff et al., 2019) and Slot-Attention (Locatello et al., 2020), the first 70K samples from the CLEVR-6 dataset and the first 60K samples from the Multi-dSprites and Tetrominoes datasets are used for training. Evaluation is performed on 320 test data examples.

**Implementation**   For the EGO model, we first encode the image input $\mathbf{x}$ using a CNN backbone. We use the same CNN architecture from Slot-Attention (Locatello et al., 2020), which is augmented with positional embeddings for all results in this section. To condition our EGO-Sum model $E(\mathbf{x}, \mathbf{z}; \boldsymbol{\theta})$ on the latent variables, we use an MLP to project $\mathbf{z}$ to a vector, replicating it to match the spatial dimension of the image features and concatenating it along the channel dimension. We process the joint features through a set of self-attention layers, followed by a global average pooling layer and a fully-connected layer to produce the final energy value. For the EGO-Attention model, we obtain the energy value from Equation 5 in Section. 2.1. We do not use dropout in our self-attention and cross-attention layers. To decode the inferred latent variables into reconstructions, we apply spatial broadcast decoding and use the same architecture from IODINE (Greff et al., 2019). We use $D_{\mathbf{z}} = 64$ for the latent variable dimension, as in baseline methods. We use $K = 7$ latent variables for CLEVR-6, $K = 6$ for Multi-dSprites, and $K = 4$ for Tetrominoes, which is one more than the maximum number of objects in the corresponding datasets. In Langevin MCMC sampling, we set the step size $\epsilon = 0.1$ and the number of Langevin steps $T = 5$. We train the model using the Adam optimizer (Kingma & Ba, 2015) with a learning rate of 0.0002 for 500K iterations, with a batch size of 128. Additional implementation details and results can be found in the Appendix.

**Segmentation accuracy**   Following the evaluation protocol in existing literature, we use the Adjusted Rand Index (ARI) (Hubert & Arabie, 1985) metric, which measures how accurately the model can decompose the scene into discrete objects. We compare our model against a variety of baseline methods, including Slot Attention (Locatello et al., 2020), IODINE (Greff et al., 2019), and MONet (Burgess et al., 2019). Similar to IODINE and Slot-Attention, we take the alpha masks generated by the spatial broadcast decoder for each latent variable, compute the clustering similarity with the ground truth masks using the ARI metric, and exclude the background in the evaluation. We report the ARI scores across different datasets in Table 1. As can be seen, EGO consistently outperforms the baselines, achieving near-perfect segmentation accuracy on CLEVR-6 and Tetrominoes, and substantially better results on Multi-dSprites compared to the existing state-of-the-art. Notably, there are more occlusions among objects presented in Multi-dSprites, which makes the task more challenging and demonstrates the effectiveness of our model in handling more complex scenes.

**Downstream prediction**   To investigate the usefulness and quality of the learned representations, We evaluate our learned object-centric model on downstream object property prediction tasks. Similar to IODINE, we probe pre-trained models' learned representations by training a linear model on top of the latent variables to predict associated object properties, such as color, shape, size, and position. Thanks to the object-centric nature of baseline methods and our model, where object representations share a common format, we can train a single probing model to independently extract properties from each object-centric latent variable. We train the probing model by using the Hungarian algorithm (Kuhn, 1955) to match the latent variables to the ground-truth objects. Following the same training and evaluation procedure described in (Dittadi et al., 2022), we compare our pre-trained EGO-Attention model against baseline approaches across different datasets in Figure 2.

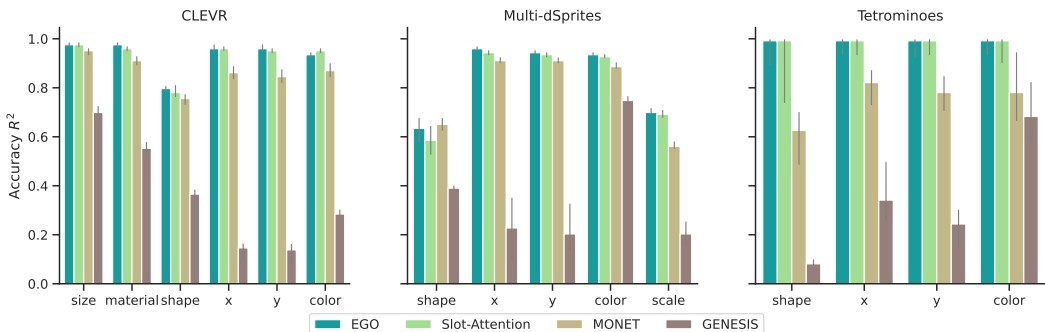

Figure 2: Downstream object property prediction results on CLEVR, Multi-dSprites, and Tetrominoes. The metric is accuracy for categorical properties or $R^2$ for numerical properties.

We see that the learned representations from our model are highly informative for predicting object properties and are comparable to or outperform the competitive baseline methods on all datasets.

## 4.2 SCENE DECOMPOSITION AND MANIPULATION

We next qualitatively study the learned object-centric representations by inspecting scene decompositions and visualizing the inference procedure. We also demonstrate the ability of our model to manipulate the scene by recomposing learned energy functions to dynamically manipulate scenes.

**Scene decomposition**  We visualize per-latent variable reconstruction results from the trained `EGO-Attention` model across different datasets in Figure 3(a). The examples show that our model is able to decompose the scene into highly-interpretable segmentations, which align well with the ground-truth objects. Extra latent variables are assigned to the background when there are more latent variables than the number of objects in the scene. Our model learns to spread objects across latent variables without explicit supervision, and object properties are also well-preserved in the inferred representations. Meanwhile, when multiple objects occlude each other, the model is able to infer the correct parts of objects by leveraging other clues such as shape and color. We further visualize scene reconstructions at each MCMC sampling step in Figure 3(c), showing that the inferred scenes are iteratively refined within a few steps. We also plot the energy function values evaluated at each Langevin dynamics iteration from a trained model with $T = 10$ steps on the Tetrominoes dataset in Figure 3(b). As can be seen, both energy values are monotonically decreasing, indicating that the model can infer the latent variables by optimizing the energy function efficiently and stably.

**Scene manipulation**  With trained `EGO` models, learned energy functions can be used to dynamically manipulate a scene's constituent objects. To show the *controllable* scene manipulation ability of `EGO`, in Figure 4(a), we show that we can combine arbitrary objects from different visual observations ($\mathbf{x}_1$ and $\mathbf{x}_2$) together into a novel scene, by sampling the latent variables from the joint EBM $E(\mathbf{x}, \mathbf{z}; \boldsymbol{\theta}) = E(\mathbf{x}_1, \mathbf{z}; \boldsymbol{\theta}) + E(\mathbf{x}_2, \mathbf{z}; \boldsymbol{\theta})$, known as product-of-experts (Hinton, 2002), and reconstructing the scene from the inferred latent variables. A visualization of the scene reconstructions and predicted masks associated with each latent variable is also included in the figure, showing that, starting from the first few iterations, the model captures object components from both images and combines them across the latent variables to generate the complete scene. We additionally show another example in Figure 4(b), where we can remove any specific object from the scene by reusing learned energy functions. As described in (Du et al., 2020), we form a new energy function as $E(\mathbf{x}, \mathbf{z}; \boldsymbol{\theta}) = E(\mathbf{x}_1, \mathbf{z}; \boldsymbol{\theta}) - E(\mathbf{x}_2, \mathbf{z}; \boldsymbol{\theta})$ to remove the objects shown in $\mathbf{x}_2$ from the scene $\mathbf{x}_1$. Similarly, we also illustrate the intermediate results at each sampling step, where we can see that in the first few steps of the sampling procedure, the model recovers the complete scene from $\mathbf{x}_1$, and then gradually removes the objects in $\mathbf{x}_2$ by optimizing the latent representations towards the region where $E(\mathbf{x}_2, \mathbf{z}; \boldsymbol{\theta})$ is higher. These results clearly demonstrate that the EBM formulation of `EGO` allows us to control the scene composition combinatorically, leading to systematic generalization to unseen object combinations.

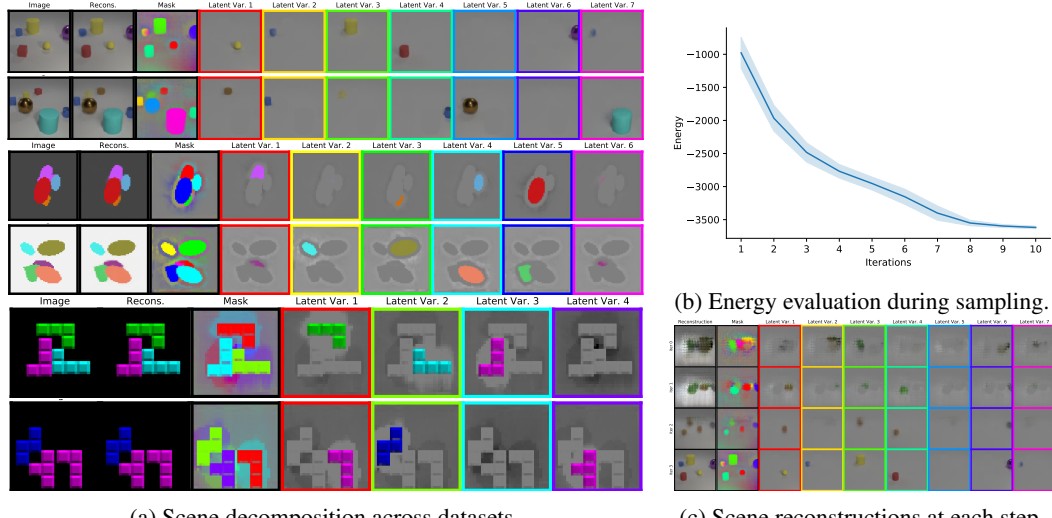

(a) Scene decomposition across datasets.

(b) Energy evaluation during sampling.

(c) Scene reconstructions at each step.

Figure 3: (**a**) Per-latent variable reconstructions and masks on CLEVR-6 (top), Multi-dSprites (middle), and Tetrominoes bottom. (**b**) The progression of energy function evaluations during Langevin sampling. (**c**) Scene decomposition and reconstruction visualization at each sampling step.

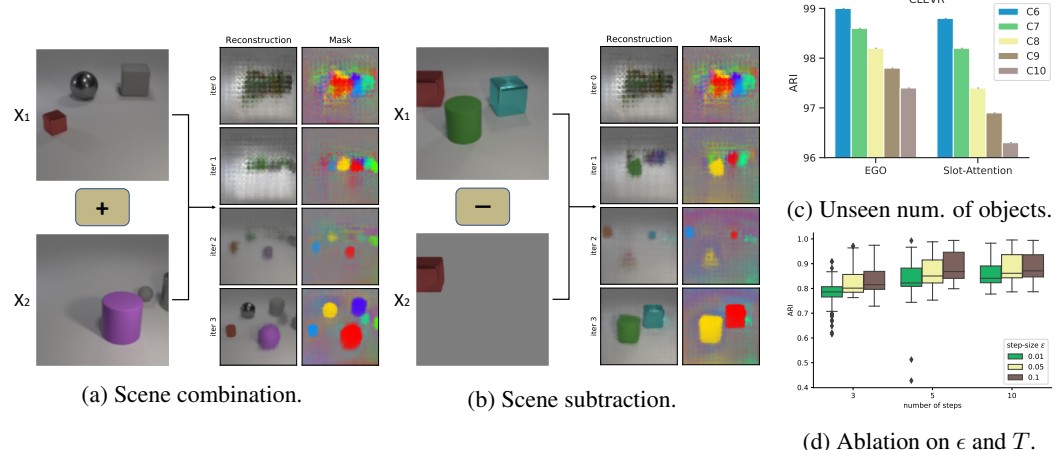

(a) Scene combination.

(b) Scene subtraction.

(c) Unseen num. of objects.

(d) Ablation on $\epsilon$ and $T$.

Figure 4: (**a**) Scene manipulation for combining objects, full version in Figure 8. (**b**) Scene manipulation for removing objects, full version in Figure 9. (**c**) ARI scores of pre-trained `EGO` on CLEVR dataset when generalizing to higher numbers of objects than seen during training. We use 'C' to denote CLEVR for brevity. (**d**) Model ablation results.

## 4.3 ROBUSTNESS AND GENERALIZATION EVALUATION

Finally, we evaluate the generalization capability of `EGO` to out-of-distribution (OOD) visual scenes and investigate its robustness with respect to model hyperparameters and component choices in an ablation study.

**Unseen object style** We first study the generalization capability of our learned object-centric representations to unseen object styles by altering the visual styles of objects presented in the test data distribution, including color, shape, and texture. Following (Dittadi et al., 2022), we apply a random color jitter transformation to the color of a random object in the scene in CLEVR and Tetrominoes. Unseen object textures are simulated by applying random neural style transfer (Gatys et al., 2015) to a random object in CLEVR and Multi-dSprites. We use the previously-trained model from the unsupervised object discovery task and evaluate it on these modified datasets to test whether the learned representations are able to generalize to unseen object styles. ARI scores of our model and other baseline methods are shown in Figure 5(a). Our model achieves strong generalization performance

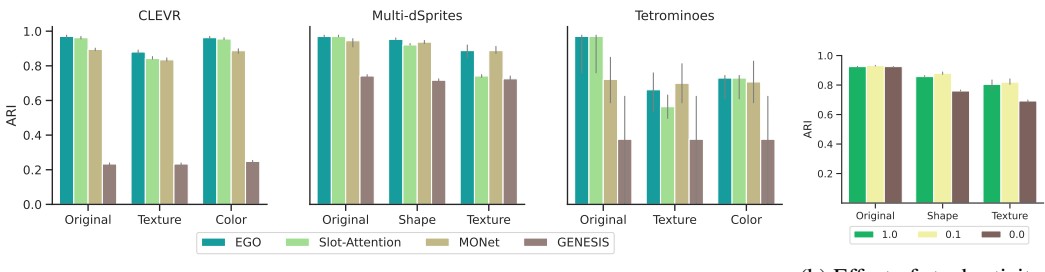

(a) ARI scores on test data with unseen object styles.

(b) Effect of stochasticity.

Figure 5: (**a**) Segmentation results of pre-trained model on OOD test dataset. (**b**) Model ablation on the effect of the noise variable in MCMC sampling.

to unseen object colors, textures, and shapes on CLEVR and Multi-dSprites, consistently providing high-quality segmentation masks in the presence of OOD object styles. We provide more details about the datasets and evaluation in the appendix.

**Increasing the number of objects**  To test how our model generalizes when more objects are present in the scene (compared to the training data), we increase the number of objects at test time. We construct a sequence of datasets by filtering CLEVR to only contain scenes with at most $[7, 8, 9, 10]$ objects, referred to as CLEVR-[7-10] respectively. We use EGO-Attention trained on CLEVR-6 with $K = 7$ latent variables, and test it on the newly-constructed datasets by increasing the number of latent variables to one more than the maximum number of objects in the corresponding dataset. We compute the ARI scores for each dataset, and report the results in Figure 4(c). We use Slot-Attention with 3 iterations (same as $T = 3$ in our model) as the baseline. As can be seen, the segmentation quality of our model remains robust to unseen numbers of objects in the test dataset, and suffers less from the increase in the number of objects than the baseline approach.

**Model ablation**  We perform ablation studies to evaluate the robustness of our approach to different hyperparameters and study the effect of different components of our model, particularly the number of MCMC sampling steps $T$, the step size $\epsilon$, and the noise in Langevin dynamics. We first run grid search over the hyperparameters, including $\epsilon \in \{0.01, 0.05, 0.1\}, T \in \{3, 5, 10\}$, number of attention blocks $\in \{1, 2, 3\}$, and dropout rate $\in \{0.0, 0.1\}$, and report the ARI scores on the Multi-dSprites dataset in Figure 4(d). Our model is quite robust to a wide range of the values of both the step size and the number of sampling steps, and we obtain near-optimal segmentation performance with each combination of the hyperparameters. We then investigate the role of the noise used in the Langevin dynamics sampling by introducing a weight term to rescale the noise variable in Equation 6. We train EGO-Attention variants with different rescaling weights $\in \{1.0, 0.1, 0.0\}$ on the Multi-dSprites dataset and evaluate their ARI scores in both the original dataset and the modified OOD dataset with unseen object styles. The results in Figure 5(b) demonstrate that the stochasticity brought by the noise in Langevin dynamics gives our model more robustness and generalization to novel scene configurations.

## 5 CONCLUSION

In this work we present EGO, a novel energy-based object-centric learning model. EGO successfully combines three essential ingredients of object-centric learning: (i) flexible energy-based models without strong assumptions on data generating process , (ii) minimal usage of potentially unexpressive specifically-designed neural modules, and (iii) explicitly modeling randomness to allow one-to-many mappings to reason about occluded or partially-observed objects. We empirically demonstrate that EGO can achieve state-of-the-art performance on various unsupervised object discovery tasks and excels at generalizing to OOD scenes. Meanwhile, controllable scene generation and manipulation are also feasible by reusing learned energy functions. These advantages make EGO a strong candidate for scaling unsupervised object-centric learning to real-world datasets, as a promising next step for future investigation.

## REPRODUCIBILITY STATEMENT

In order to make the experimental results easily reproducible, we made following efforts: i) Source code necessary to reproduce results of our model is made available in the supplementary material. ii) We provided the detailed training procedure in Algorithm 1, and described all model architectures and hyperparameters in main text (Section 4) and appendix (Section A).

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

## A  Implementation Details

### A.1  Datasets and Preprocessing

We describe more details about the datasets used in our experiments, and the preprocessing steps we applied during training and evaluation.

**CLEVR**  We use the CLEVR (Johnson et al., 2017) dataset from the Multi-Object Datasets library (Kabra et al., 2019)[1]. Each data example in the CLEVR dataset contains a rendered image of a scene with a set of up to 10 objects. The multiple objects in the scene can possibly occlude each other, and each object can be rendered with different shapes (e.g., cube, sphere, cylinder), colors (Red, Cyan, Green, Blue, Brown, Gray, Purple, Yello), sizes (small, large), materials (rubber, metal), and positions ($x$ and $y$ coordinates). Following (Greff et al., 2019; Locatello et al., 2020), a center crop of the image followed by resizing to $128 \times 128$ is applied first, then we transform the RGB values to the range $[-1, 1]$.

**Multi-dSprites**  We use the Multi-dSprites dataset from the dSprites (Matthey et al., 2017) dataset. More specifically, we use the variant with grayscale background and colored sprites as done in (Greff et al., 2019; Locatello et al., 2020). Each data example in the Multi-dSprites dataset contains a scene with up to 5 objects, and each object can be rendered with different shapes (ellipse, square, hear), colors (HSV space), scales (in $[0.5, 1]$), positions ($x$ and $y$ coordinates, in $[0, 1]$) and orientations. Following (Greff et al., 2019; Locatello et al., 2020), we keep the resolution at $64 \times 64$ and transform the RGB values to the range $[-1, 1]$.

**Tetrominoes**  We use the Tetrominoes dataset from the Multi-Object Datasets library (Kabra et al., 2019). Each data example in the Tetrominoes dataset contains a scene with exactly 3 tetris pieces with a black background. The tetris pieces can be rendered with different shapes (19 different shapes), colors (Yellow, Purple, Red, Blue, Green, Cyan), and positions ($x$ and $y$ coordinates). Following (Greff et al., 2019; Locatello et al., 2020), we keep the resolution at $35 \times 35$ and transform the RGB values to the range $[-1, 1]$.

**OOD dataset variants**  For the evaluation of our model on constructed OOD datasets with unseen object styles and colors, we use the library[2] from the benchmark (Dittadi et al., 2022). To apply the random color transformation to the data examples, a color jittering `torchvision.transforms.ColorJitter(brightness=0.5, contrast=0.5, saturation=0.5, hue=0.5)` is applied to a randomly selected object in the scene. To apply the random style transformation, neural style transfer is used to transfer the first 100K samples from all datasets. Please refer to Figure.11 in (Dittadi et al., 2022) for qualitative evaluation of the introduced visual effects of object style and color. For more detailed descriptions of the transformations used for constructing the OOD dataset variants, please refer to (Dittadi et al., 2022).

### A.2  EGO Model Architecture

We introduced the detailed neural network architecture of our model used in our experiments here.

**Image Encoder**  We use the same CNN backbone in the Slot-Attention work (Locatello et al., 2020) to encode the visual scene input into a spatial feature map. For CLEVR dataset, the CNN network consists of 4 convolutional layers with $5 \times 5$ kernel size, $[64, 64, 64, 64]$ channels, $[1, 1, 1, 1]$ zero-padding size respectively, and each convolutional layer is followed by a ReLU activation function. Positional encoding is applied to the encoded feature map to provide the representation with positional information for better modeling the consistency between images and latent variables. More details about the CNN backbone architecture on Multi-dSprites and Tetrominoes datasets can be found in (Locatello et al., 2020).

---

[1]`https://github.com/deepmind/multi_object_datasets`
[2]`https://github.com/addtt/object-centric-library`

**EGO Module**   As illustrated in Figure 1, our model consists of several vanilla attention blocks. For CLEVR and Multi-dSprites datasets, we use $L = 3$ cross-attention blocks, $T = 3$ sampling steps, $\epsilon = 0.1$ step size and $K = 7$ latent variables. We use 1 attention head, and do not use any dropout in the attention layer or the MLP layer. For Tetrominoes dataset, we found using $L = 1$ cross-attention block would also lead to near-perfect results. We optionally learn the parameters of the initial Gaussian distribution of $\mathbf{z}^0$, and found it helps in Tetrominoes dataset. We use $D_{\mathbf{z}} = 64$ for all models across datasets, and all MLP layers have the same number of hidden units with ReLU activation function.

**Image Decoder**   We use the spatial broadcast decoder to decode the inferred latent variables into scene reconstruction. The decoder consists of several deconvolutional layers, augmented with positional encoding. For detailed description of the deconvolutional layers, please refer to (Locatello et al., 2020).

### A.3   MODEL TRAINING AND EVALUATION

We implemented our model in Jax (Bradbury et al., 2018) and Flax (Heek et al., 2020). We train all models with batch size 128 using the Adam optimizer (Kingma & Ba, 2015) with a learning rate 0.0002 for 500K steps. We use the cosine learning rate schedule (Loshchilov & Hutter, 2016) with warmup steps 2500. We clip the gradients to maximum global norm 1, though we generally found that the training process is quite stable without gradient clipping. We train our models on 8 Nvidia A100 GPUs, and the training time on CLEVR dataset is about 1 day, and within a few hours on Multi-dSprites and Tetrominoes datasets.

## B   ADDITIONAL EXPERIMENTS

We provide additional experimental results on scene decomposition, scene manipulation, and model ablation.

### B.1   SCENE DECOMPOSITION

We illustrate more examples of visualizing the scene decomposition and reconstruction results at each sampling step across datasets in Figure 6, 7, accompanying the results shown in Figure 3.

We additionally consider visualizing the per-step decomposition results in a more extreme case when we have much more sampling steps $T$. In doing so, we train a model with $T = 10$ on Tetrominoes dataset and visualize the per-iteration reconstruction results in Figure 13. We can see from the figure that after 6 iterations, the model already reconstructed the almost complete scene, which can be seen as qualitative evidence for the sufficiency of using a small number of sampling steps in the latent space.

Besides the ARI scores for evaluating the segmentation performance in unsupervised object discovery task, we additionally evaluate our model in terms of other metrics, mean squared error in reconstruction (MSE) and mean segmentation covering (mSC). Detailed description of the metrics are introduced in (Dittadi et al., 2022). We report the results in Figure. 10. We see that EGO achieves comparable performance in terms of the reconstruction quality compared to other methods, and the segmentation accuracy in terms of mSC remains across different datasets are consistent with the observation on ARI scores, where EGO shows competative or superior performance.

### B.2   SCENE MANIPULATION

We also show the per-iteration reconstruction results for the examples shown in Figure 4(a,b). In the first few iterations, the model recovers the global scene and then seeks for the updates to satisfy the constraints of latent representation given by the joint energy function.

### B.3   MULTI-MODALITY AND MULTI-STABILITY

We present qualitative visualization of the multi-modal posterior in EGO. Similar to (Greff et al., 2019), we construct a visual scene input with multiple possible explanations, and display the per-

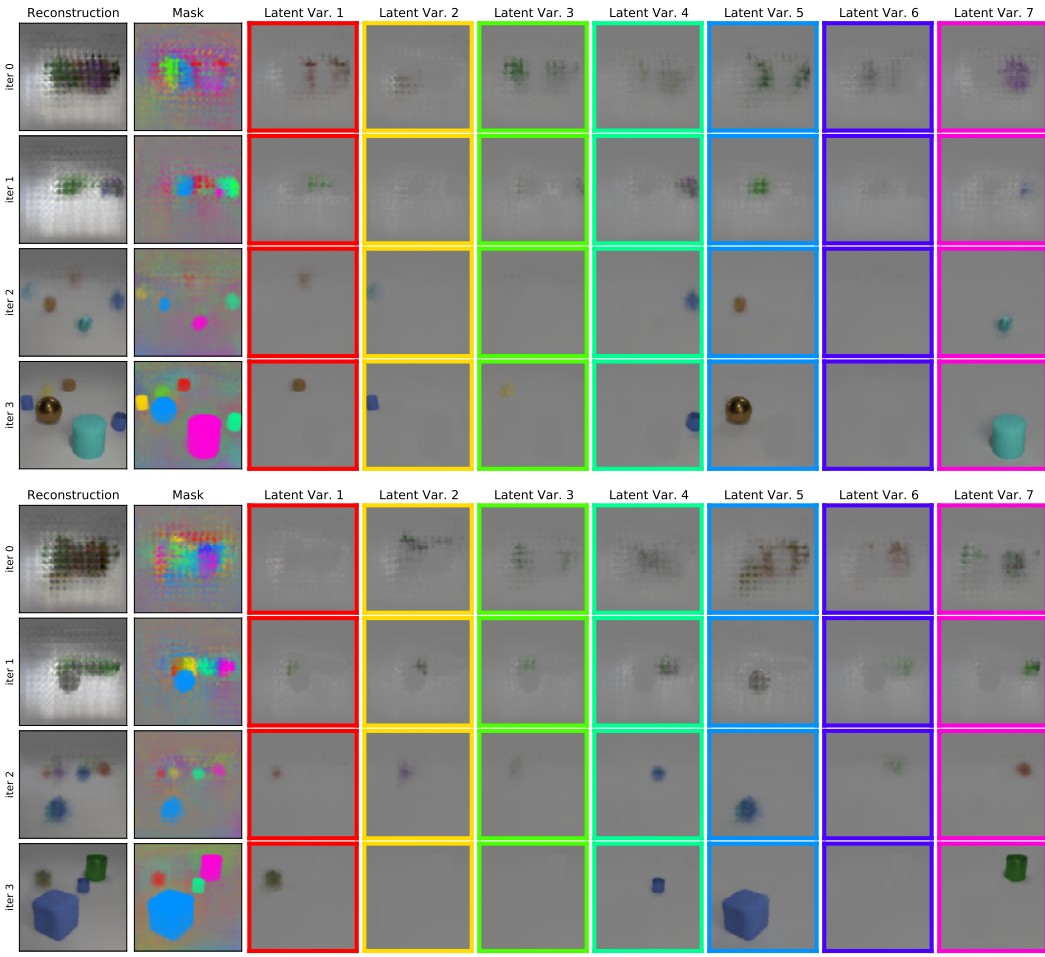

Figure 6: Scene decomposition and reconstruction results at each sampling step on CLEVR.

latent variable reconstructions in Figure. 11. We can see that without distinguishable clues like color and position, we can sample a set of different segmentation configurations to explain the ambiguous visual scene input, showing the multi-modality in the learned posterior.

## B.4   MODEL ABLATION

We include more model ablation results, studying the effects of magnitude of noise in sampling, number of attention blocks $L$, dropout rate in `EGO` MLP, dropout rate in `EGO` attention layer, and gradient clipping norm on Tetrominoes. We plot the distribution of the ARI scores for each configuration of the considered hyperparameters, grouped by the number of sampling steps $T$ in $x$-axis. We show the results in Figure 12. As can seen from the results, our proposed model is robust to almost any configurations of the hyperparameters, where near-optimal performance can be attained in each setup.

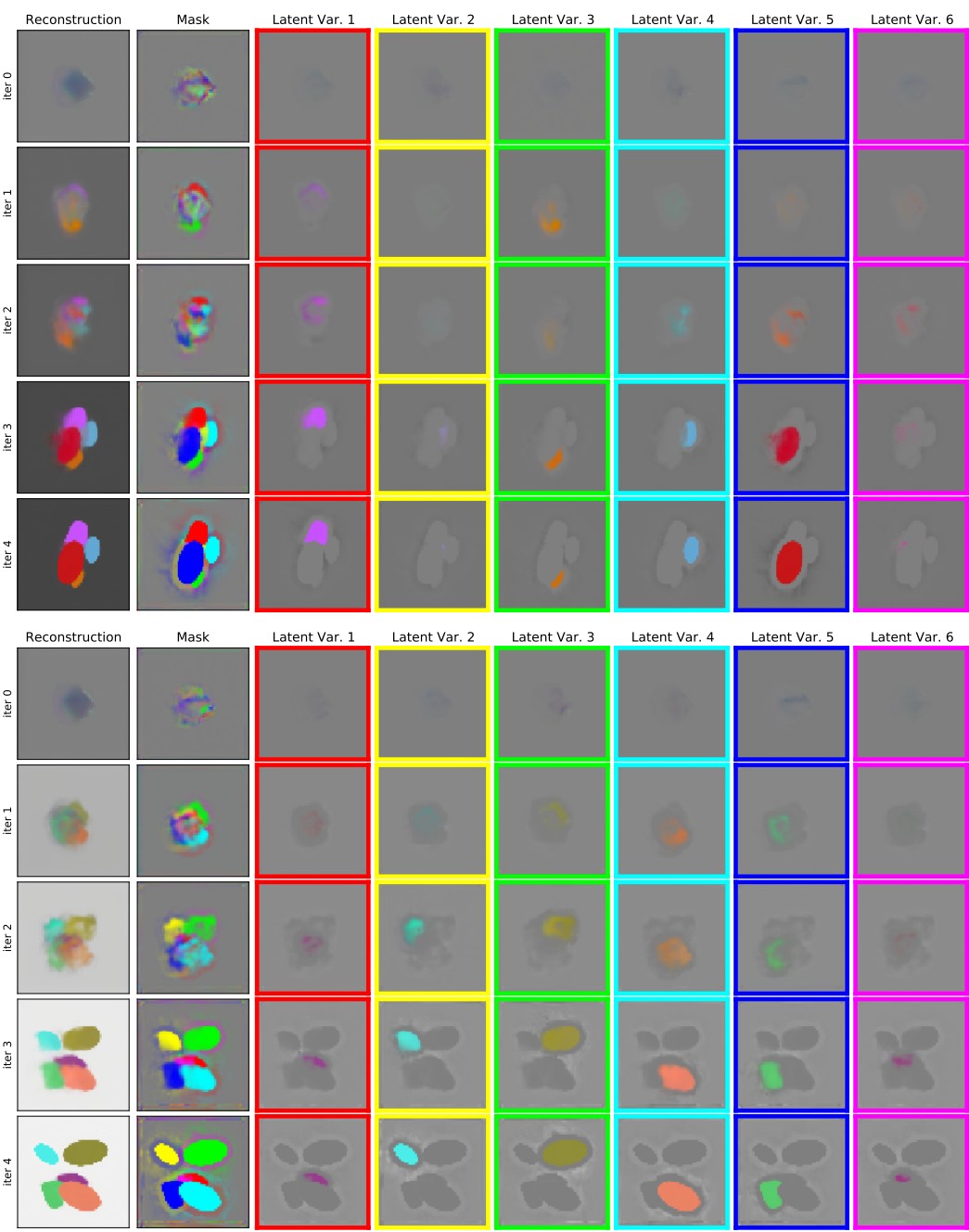

Figure 7: Scene decomposition and reconstruction results at each sampling step on Multi-dSprites.

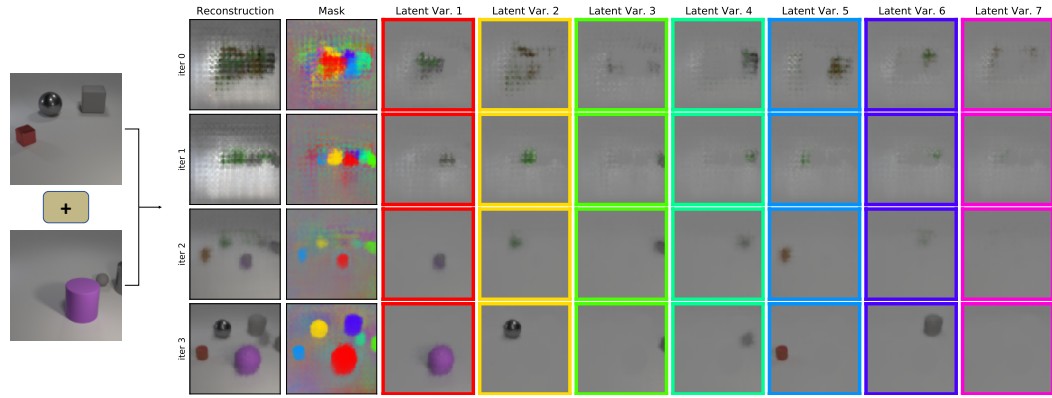

Figure 8: Scene manipulation: Combining objects from different images into a novel scene using learned energy function.

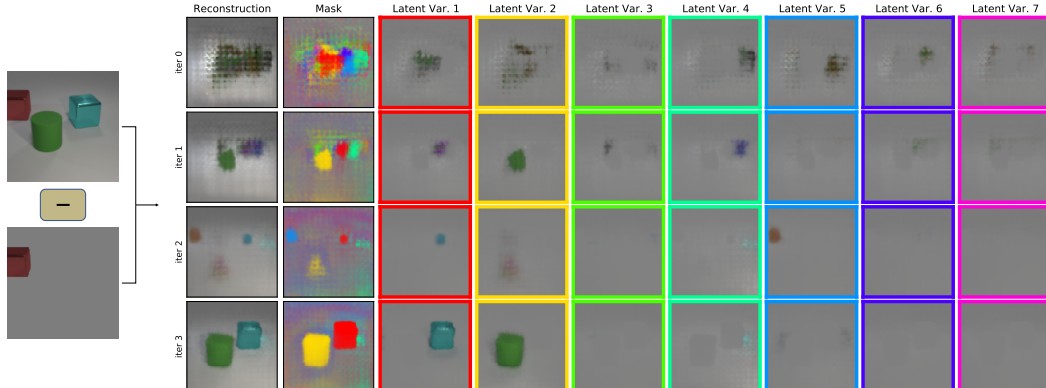

Figure 9: Scene manipulation: Removing specific object from a scene using learned energy function.

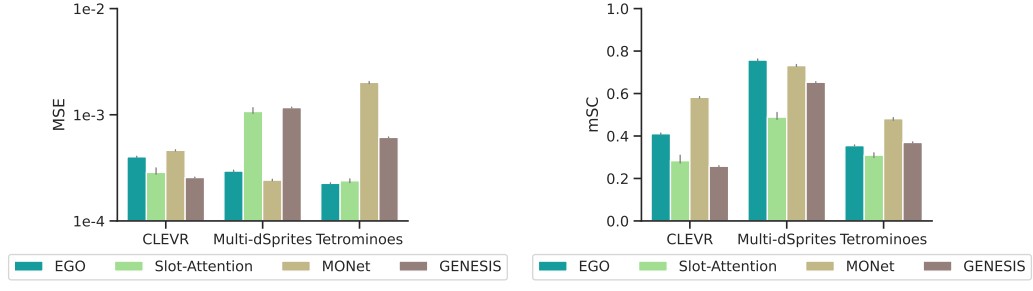

Figure 10: Evaluation of reconstruction quality in MSE and segmentation accuracy in mSC.

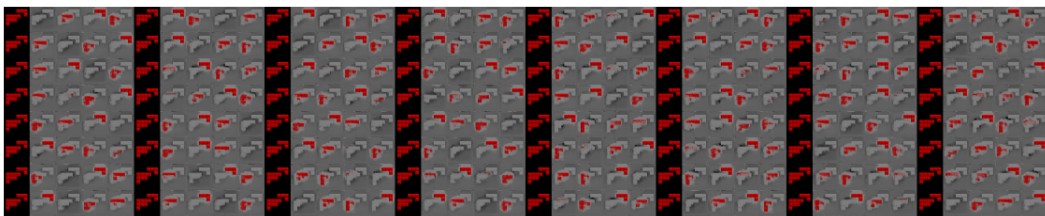

Figure 11: Visualization of samples from multi-modal posterior from EGO on Tetrominoes.

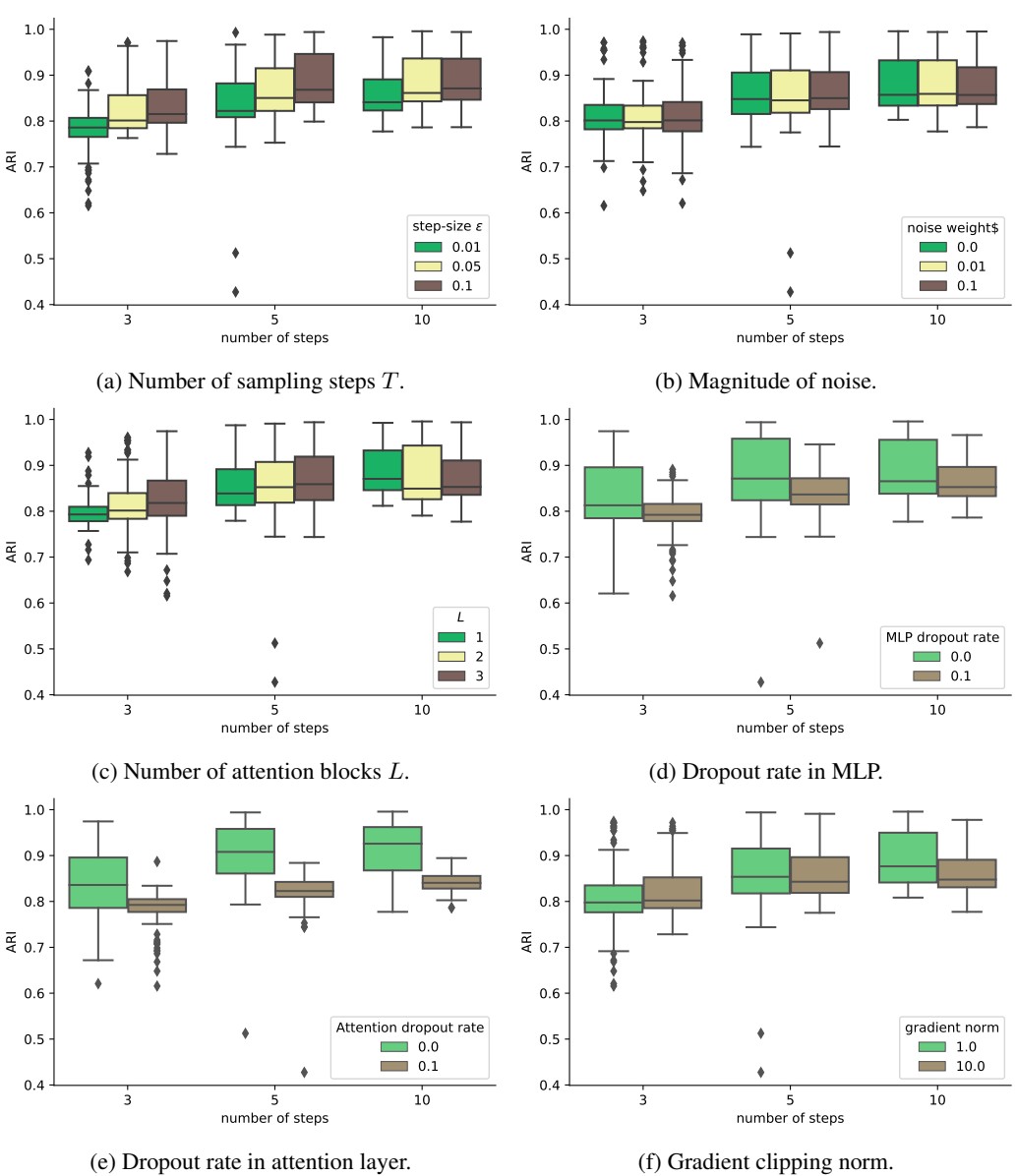

(a) Number of sampling steps $T$.

(b) Magnitude of noise.

(c) Number of attention blocks $L$.

(d) Dropout rate in MLP.

(e) Dropout rate in attention layer.

(f) Gradient clipping norm.

Figure 12: Model ablation results: ARI scores by varying individual hyperparameter choices, grouped by the number of sampling steps.

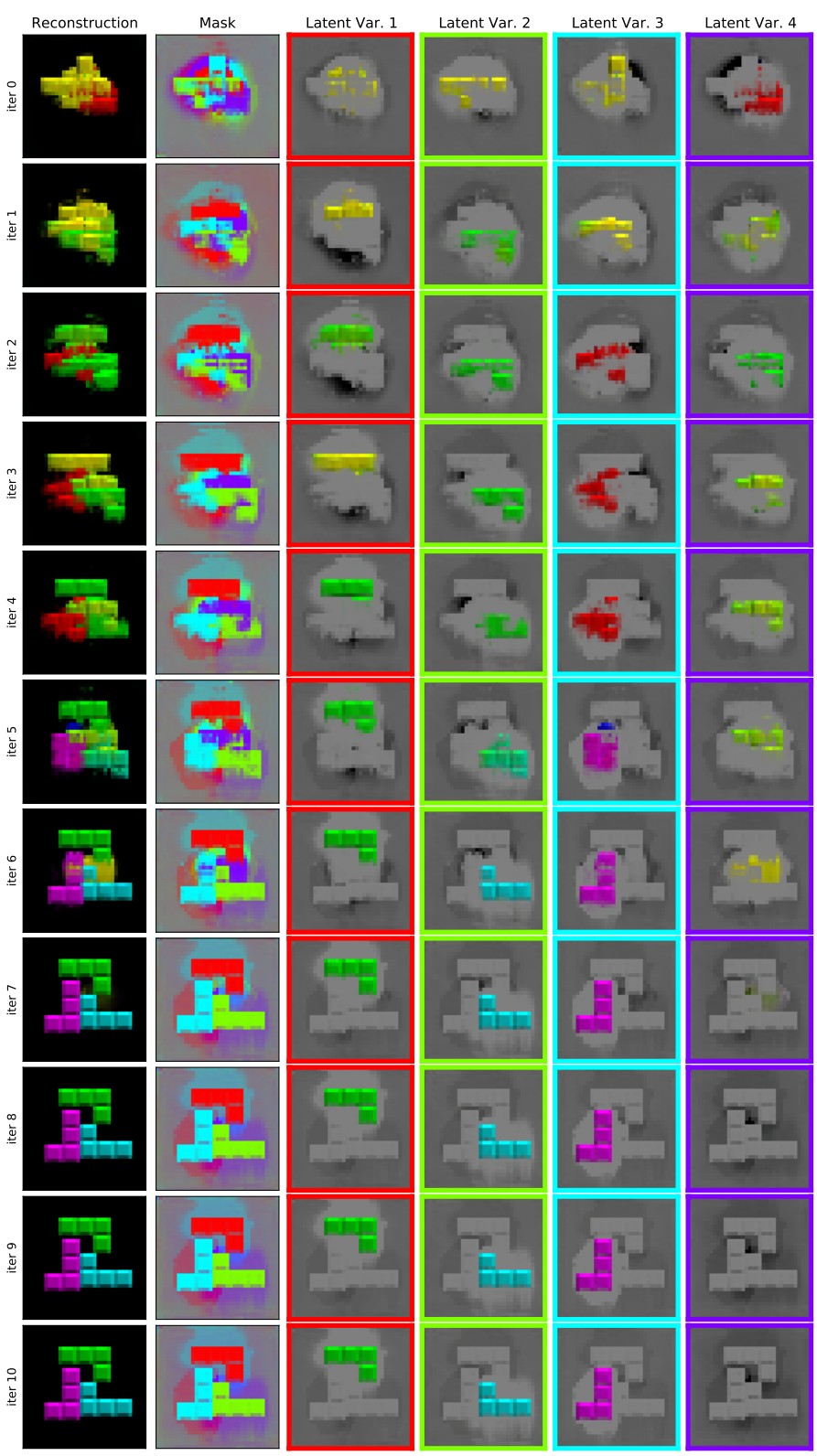

Figure 13: Scene decomposition and reconstruction results at each sampling step on Tetrominoes.

