# OpenReview forum: "Robust and Controllable Object-Centric Learning through Energy-based Models"
_ICLR.cc/2023/Conference — ICLR 2023 poster_

### Official Review · Reviewer_vs6t · 2022-10-20

**Confidence:** 5
**Correctness:** 3
**Technical Novelty And Significance:** 3
**Empirical Novelty And Significance:** Not applicable
**Recommendation:** 6

**Clarity, Quality, Novelty And Reproducibility:**

The paper is written clearly and appears to be reproducible. However, overall the main contribution (EGO) appears to be incremental when taken in the context of the existing body of work.

**Strength And Weaknesses:**

Strengths
========
The core idea, which I believe is casting slot inference as sampling slots consistent with an image as scored by an energy function, seems to be a novel and interesting approach to this task.

The presentation of the paper is solid.

Weaknesses
========
I have concerns with various claims made throughout the paper.

- **[EGO is…] “conceptually simple”/“Without the need for specially-tailored neural network architectures”** This is provided as one of the main contributions of the paper. I disagree that EGO is any simpler than the key baseline, Slot Attention. The “variant” of EGO explored in this paper (Algorithm 1) and Slot Attention are fairly similar methods in terms of the neural modules employed and the use of an iterative inference algorithm.

- **“Without the need for … excessive generative modeling” / “Minimal assumptions on the generative process”** When comparing EGO with previous generative object-centric models like GENESIS and GENESIS-v2 [1], it is important to consider that these models support unconditional generation of novel scenes because they have a prior distribution over slots. EGO sacrifices this capability in favor of a simpler approach to slot inference. Also, Slot Attention, the most relevant baseline, does not make any generative modeling assumptions either.

- **“We show that EGO can be easily integrated into existing architectures”** This is also provided as one of the main contributions of EGO, but it is not actually shown outside of the specific encoder-decoder object representation learning context. No other examples of how EGO can be used are provided. In fact, EGO's ability to discover objects seems to hinge on the use of the spatial broadcasting decoder (SBD) as an inductive bias to encourage each latent slot to fixate on a distinct scene object, in the same manner as the baselines (see [2]). Demonstrating how EGO can be used in a variety of diverse contexts to learn slots that fixate, without supervision, on objects would be one way to improve the paper.

I struggle to see the novel conceptual insights or new capabilities enabled by EGO. One supposedly new capability --- compositional scene editing --- is explored in the experiments, but I believe the baseline object-centric methods can also support this to some degree? It would be good to compare against the performance of at least the key baseline, Slot Attention, at this. Also, I believe IODINE is capable learning a multi-modal distribution over slots (like EGO)? This capability is not empirically explored in the paper, however.

Issues of instability [3] and difficulty with more realistic scenes [4] for object-centric representation learning methods (particularly those based on the SBD) have been demonstrated in the literature, yet these issues are not discussed in this work. I could not find any discussed limitations.

Overall, the segmentation results seem to be comparable to Slot Attention, with only marginal demonstrated improvements in segmentation on Multi-dSprites. Perhaps conducting experiments on more challenging datasets would reveal a larger gap in performance with respect to baselines?

References
========
1. Engelcke, Martin, Oiwi Parker Jones, and Ingmar Posner. "Genesis-v2: Inferring unordered object representations without iterative refinement." Advances in Neural Information Processing Systems 34 (2021): 8085-8094.
2. Singh, Gautam, Fei Deng, and Sungjin Ahn. "Illiterate dall-e learns to compose." International Conference on Learning Representations. 2021.
3. Chang, Michael, Thomas L. Griffiths, and Sergey Levine. "Object representations as fixed points: Training iterative refinement algorithms with implicit differentiation." arXiv preprint arXiv:2207.00787 (2022).
4. Karazija, Laurynas, Iro Laina, and Christian Rupprecht. "Clevrtex: A texture-rich benchmark for unsupervised multi-object segmentation." arXiv preprint arXiv:2111.10265 (2021).

**Summary Of The Paper:**

This paper introduces a EGO, a neural module for object-centric representation learning. EGO casts the inference of object-centric latent variables as energy-based modeling. It uses Langevin MCMC to sample latent variables consistent with the given input image. Simplicity of the neural network architecture is emphasized. Empirical evidence of the validity of the approach is provided in the form of segmentation and property prediction on the Multi-Object Datasets environments. Additional experiments evaluating hyperparameter choices, robustness, and generalization are provided as well. Overall, the performance of EGO appears to be comparable to the most similar baseline Slot Attention.


**Summary Of The Review:**

Overall, I believe this work is not yet ready for publication due to various incorrect/unjustified claims and limited novelty of the contributions (as described under Weaknesses). More justification and evidence is needed as to A) why this EBM-based approach constitutes a promising new direction for object-centric representation learning and B) whether EGO is truly a generic module for learning object-centric representations in other architectures/contexts as claimed. I believe with these improvements, the work could be a valuable contribution.

---
Update after rebuttal: I have increased my score from 3 --> 6 to reflect my new perspective of this work after discussion and improvements. See comment thread below for more details.

---

> ### Author Response · Authors · 2022-11-19
> **Response to reviewer vs6t (part 2)**
>
>
> > **Q3** “We show that EGO can be easily integrated into existing architectures” This is also provided as one of the main contributions of EGO, but it is not actually shown outside of the specific encoder-decoder object representation learning context. No other examples of how EGO can be used are provided. In fact, EGO's ability to discover objects seems to hinge on the use of the spatial broadcasting decoder (SBD) as an inductive bias to encourage each latent slot to fixate on a distinct scene object, in the same manner as the baselines (see [2]). Demonstrating how EGO can be used in a variety of diverse contexts to learn slots that fixate, without supervision, on objects would be one way to improve the paper.
>
> 1. EGO can be easily integrated into existing architectures, since it can be wrapped up as a stand-alone module and plugged into the task-specific architecture, like the slot-attention module.
> 2. In the context of unsupervised object discovery, every baseline approach, as well as EGO, rely on reconstruction-based loss, which further needs to be combined with some form of decoders. In our case, we use the spatial broadcasting decoder to align with existing works, such that we can isolate the effects of EGO and make fair comparison against baseline methods. Therefore, we believe that we cannot connect the use of the encoder-decoder architecture to the concern of flexibility in integrating EGO into existing architectures.
> 3. To further support our claim that EGO can be easily integrated into task-specific models, we conducted additional experiments on set prediction task on CLEVR6 dataset, by replacing slot-attention module with EGO. In this experiment, we no longer have spatial-broadcasting decoder, and use a weight-shared MLP classification head to predict the targets. We show the preliminary experimental results in the table below, where we can see that EGO outperforms slot-attention in terms of the AP metrics.
>
>   | Model | AP$_\infty$ | AP$_1$ | AP$_{0.5}$ | AP$_{0.25}$ | AP$_{0.125}$ |
>   | --- | --- | --- | --- | --- | --- |
>   | Slot MLP | 19.8 | 1.4 | 0.3 | 0.0 | 0.0|
>   | DSPN | 85.2 | 81.1 | 47.4 | 10.8 | 0.6 |
>   | Slot-Attention | 94.3 | 86.7 | 56.0 | 10.8 | 0.9 |
>   | EGO | 95.2 | 88.5 | 58.1 | 11.4 | 1.1 |
>
> > **Q4** One supposedly new capability --- compositional scene editing --- is explored in the experiments, but I believe the baseline object-centric methods can also support this to some degree? It would be good to compare against the performance of at least the key baseline, Slot Attention, at this.
>
> In the scene manipulation task, we demonstrated that we can generate novel scenes by sampling from the recomposed energy functions given two scene inputs. This is a fully automatic procedure without any manual intervention and inspection. We believe that this is a new capability of EGO that is not supported by slot-attention baseline.
>
> > **Q5** I believe IODINE is capable learning a multi-modal distribution over slots (like EGO)? This capability is not empirically explored in the paper, however.
>
>
>
> > **Q6** Issues of instability [3] and difficulty with more realistic scenes [4] for object-centric representation learning methods (particularly those based on the SBD) have been demonstrated in the literature, yet these issues are not discussed in this work. I could not find any discussed limitations.
>
> 1. We did not encounter any serious instability issue in our experiments. We conducted extensive ablation studies on various hyperparameters and found that EGO is quite robust to the choice of hyperparameters within the default range.
> 2. We agree that more discussion on the potential limitations of EGO is needed. We have added more discussion on this in the conclusion section of the revised manuscript.
>
> > **Q7** Perhaps conducting experiments on more challenging datasets would reveal a larger gap in performance with respect to baselines?
>
> For the experimental consideration of datasets, we follow the same setting in slot-attention for better comparison against the baseline methods. Our methods can achieve comparable performance on the simpler datasets (CLEVR6 and Tetrominoes) where the ARI scores are nearly perfect. We show that our approach outperforms all baseline methods on the more challenging Multi-dSprites dataset, where more occlusion among objects is present. Besides the regular evaluation of the segmentation performance, we also conducted an extensive evaluation of the robustness and generalization to out-of-distribution scenes. We show that our model is consistently superior to the baseline methods when facing unseen object styles or number of objects, in terms of both segmentation performance and downstream prediction.

---

> > ### Comment · Reviewer_vs6t · 2022-11-29
> > **Thanks**
> >
> > Dear authors,
> >
> > Thanks for responding carefully to my feedback. I have some follow-up comments.
> >
> > > Q1: our point is that both cross-attention and MCMC are well-understood so combining them naively as is done in EGO is conceptually simpler than inventing a novel attention module and designing an equivariant update rule by hand
> >
> > Thanks for the clarification. I see your point.
> >
> > > Q2: ...Energy-based models in our work show advantages over the slot-attention approach on: i) **scene generation**, where we show that we can recompose learned energy-functions to generate novel scenes...
> >
> > It would help to clarify that EGO only supports **conditional** scene generation. This would make it clear that EGO sacrifices the ability to support **unconditional** scene generation in favor of simplicity. The latter is an important and separate problem.
> >
> > > Q6: We agree that more discussion on the potential limitations of EGO is needed. We have added more discussion on this in the conclusion section of the revised manuscript.
> >
> > Maybe I missed something but I do not see this in the revision. I am particularly curious about the computation (memory requirements and wall-clock time for training/testing the model) compared to Slot Attention. The added burden of computing a gradient `L x T` times seems to be significant.

---

> > > ### Author Response · Authors · 2022-11-30
> > > **Thank you for the feedback**
> > >
> > > Thank you for your follow-up questions, we appreciate your time and effort in providing us with useful feedback.
> > >
> > > > It would help to clarify that EGO only supports conditional scene generation. This would make it clear that EGO sacrifices the ability to support unconditional scene generation in favor of simplicity. The latter is an important and separate problem.
> > >
> > > To clarify this point, we will add a sentence to the _Object-centric learning_ paragraph in the _Related Work_ section, to explicitly state the trade-offs made by EGO and its limitations on unconditional scene generation, as follows:
> > >
> > > ""Different from works like GENESIS (Engelcke et al., 2020; 2021) which has built-in support for unconditional scene generation, EGO does not assume a parametric prior distribution over latent variables in favor of a simple approach to object-centric inference and conditional scene manipulation with learned energy functions.""
> > >
> > > > I am particularly curious about the computation (memory requirements and wall-clock time for training/testing the model) compared to Slot Attention. The added burden of computing a gradient L x T times seems to be significant.
> > >
> > > We discussed the wall-clock running time statistics of EGO briefly in Section A.3 in the appendix, where we show that in practice, our method has comparable training time with other baseline methods. For memory consumption, we can train the model with a batch size of 32 on a single 32GB V100 GPU on the CLEVR dataset, which is more than enough for practical use and comparable to other methods. Though the MCMC sampling inference involves extra gradient computations, we find that the computational cost is negligible, mostly thanks to i) the efficient implementation via jax.vmap and XLA, and ii) the fact that both $L$ and $T$ are small in all our models, which are both set to $3$.

---

> > > > ### Comment · Reviewer_vs6t · 2022-12-04
> > > > **Reply**
> > > >
> > > > > To clarify this point, we will add a sentence to the Object-centric learning paragraph
> > > >
> > > > Thanks!
> > > >
> > > > > Though the MCMC sampling inference involves extra gradient computations, we find that the computational cost is negligible, mostly thanks to i) the efficient implementation via jax.vmap and XLA, and ii) the fact that both and are small in all our models, which are both set to 3
> > > >
> > > > A requirement of using jax.vmap and XLA to make this method practical is concerning. Also, in Section 4.1 under "Implementation" the paper currently says EGO uses T = 5 Langevin steps. Then, in Section 4.3 under "increasing the number of objects", the paper says "same as T = 3 in our model". Is the latter a typo? (should be "L = 3")? Also, according to Figures 3b and 4d, using T = 3 and L = 3 appears to achieve much worse results than higher values (in particular, for larger values of L).
> > > >
> > > > ---
> > > > Overall, the majority of my concerns with the framing of the work and the motivations have been resolved. However, I still believe there are outstanding weaknesses. In particular, while EGO suggests an alternative EBM-based paradigm for object-centric representation learning, I do not believe the limitations of the method are properly discussed. I believe the computational challenges are more severe than the paper currently makes them out to be, which may limit the viability of this proposed alternative paradigm. I hope the authors discuss this limitation more prominently in an updated version of the work. I will increase my score to reflect this perspective.

---

> > > > > ### Author Response · Authors · 2022-12-08
> > > > > **Thank you for the feedback**
> > > > >
> > > > > Thank you for your follow-up questions and valuable comments. We are happy to hear that most of your concerns have been addressed, and we greatly appreciate that you will increase the score. Please find our responses to your questions below. We hope that they can help clarify any remaining concerns and improve the evaluation to a positive one.
> > > > >
> > > > >
> > > > > > in Section 4.1 under "Implementation" the paper currently says EGO uses T = 5 Langevin steps. Then, in Section 4.3 under "increasing the number of objects", the paper says "same as T = 3 in our model". Is the latter a typo? (should be "L = 3")?
> > > > >
> > > > > Thank you for your thorough reading. In Section 4.1, we use $T=5$ as the default setting because it leads to more robust performance compared to other hyperparameters (as shown in Figure. 12(a) ), such as the noise scale $\eta$. However, our experimental results show that $T=3$ is sufficient for achieving comparable performance. In Section 4.3, we use $T=3$ to align with the setting used for studying the effects of increasing the number of objects in the slot-attention work (as shown in Figure 5 in [1]).
> > > > >
> > > > > [1] Object-Centric Learning with Slot Attention
> > > > >
> > > > > > according to Figures 3b and 4d, using T = 3 and L = 3 appears to achieve much worse results than higher values (in particular, for larger values of L).
> > > > >
> > > > > We would like to clarify that Figure 3b shows the progression of energy function values during the MCMC sampling procedure, rather than the segmentation performance. The label iterations on the x-axis indicates the time step in Langevin dynamics, which is different from $T$ in EGO, which represents the total number of MCMC steps.
> > > > > Similarly, Figure 4d primarily illustrates the effect of the step size, grouped by $T$.
> > > > > We conducted an extensive ablation study on the effects of $T$ and $L$ in Figure 12 in the appendix, which shows that the optimal performance found by grid search when $T=3, L=3$ is comparable to the best performance across the entire hyperparameter space.
> > > > >
> > > > > > A requirement of using jax.vmap and XLA to make this method practical is concerning.
> > > > >
> > > > > > I believe the computational challenges are more severe than the paper currently makes them out to be, which may limit the viability of this proposed alternative paradigm. I hope the authors discuss this limitation more prominently in an updated version of the work.
> > > > >
> > > > > We would like to highlight that the bi-level optimization procedure used in EGO is in fact quite common in many existing works and tasks, such as meta-learning and DSPN. As a result, the EGO model and implementation are not tied to any particular deep learning framework. In our own implementation, we utilized jax.vmap and XLA to enhance computational efficiency, but we do not expect their use to impair the applicability and efficiency of EGO.
> > > > > We will also add the above discussion to the revised manuscript alongside the discussion on the computational cost of EGO.

---

> ### Author Response · Authors · 2022-11-19
> **Response to reviewer vs6t (part 1)**
>
> Thank you for your detailed feedback, which we believe will improve our final manuscript. We appreciate your constructive comments and we are glad to hear that you found our work novel and interesting. We address each of the points in your review in turn, and have revised the manuscript with the corresponding clarifications:
>
> > **Q1** [EGO is…] “conceptually simple”/“Without the need for specially-tailored neural network architectures” This is provided as one of the main contributions of the paper. I disagree that EGO is any simpler than the key baseline, Slot Attention. The “variant” of EGO explored in this paper (Algorithm 1) and Slot Attention are fairly similar methods in terms of the neural modules employed and the use of an iterative inference algorithm.
>
> By "conceptually simple", we mean:
> 1. In terms of architecture, EGO does not need to invent any new neural network module, where we adopt vanilla cross-attention, while slot-attention is a tailor-made module by inventing an inverted attention mechanism to explicitly introduce competition among slots. Also, the neural architecture (an off-the-shelf cross-attention module) of EGO is much simpler than slot-attention, which mixes a special attention mechanism with recurrent neural networks (This architectural design was non-trivial, and it was a main novelty of the slot attention paper).
> 2. In terms of the update rule, EGO does not require designing an explicit update rule to perform iterative inference.  Instead, EGO relies on existing gradient-based MCMC methods, which are again already well-established in the literature.
>
> Although we understand the reviewer's view that one can question if cross-attention + MCMC is simpler than slot attention when we talk about the final form of the whole model, our point is that both cross-attention and MCMC are well-understood so combining them naively as is done in EGO is conceptually simpler than inventing a novel attention module and designing an equivariant update rule by hand.
>
> > **Q2** “Without the need for … excessive generative modeling” / “Minimal assumptions on the generative process” When comparing EGO with previous generative object-centric models like GENESIS and GENESIS-v2 [1], it is important to consider that these models support unconditional generation of novel scenes because they have a prior distribution over slots. EGO sacrifices this capability in favor of a simpler approach to slot inference. Also, Slot Attention, the most relevant baseline, does not make any generative modeling assumptions either.
>
> We agree with the reviewer that EGO does not support unconditional generation and slot-attention does not make any generative modeling assumptions either. We discuss the motivations and advantages of our (energy-based) model formulation below:
> 1. Energy-based models are less restrictive in functional form, compared to other VAE-based models (e.g. MONet, IODINE) where specific assumptions about data generating process must be explicitly made, usually in the form of directed graphical model with tractable likelihood and posterior. The choice of energy-based models offers us much more flexibility in the neural architectures, and we show that with MCMC sampling, we can further avoid the need of additional inference network in VAE-based models which needs extra engineering and algorithmic design efforts.
> 2. Energy-based models in our work show advantages over the slot-attention approach on: i) scene generation, where we show that we can recompose learned energy-functions to generate novel scenes, and ii) model robustness, since we are using stochastic MCMC-based sampler for inference while the slot-attention approach is deterministic, we show that our model is more robust to out-of-distribution samples and can achieve better segmentation performance in the presence of occlusion, compared to the slot-attention approach.
>
> To avoid potential confusion on the description of previous works, we have also revised the text to make the context clearer.

---

### Official Review · Reviewer_tTco · 2022-10-22

**Confidence:** 4
**Correctness:** 3
**Technical Novelty And Significance:** 3
**Empirical Novelty And Significance:** 3
**Recommendation:** 6

**Clarity, Quality, Novelty And Reproducibility:**

Paper writing is clear and easy to follow.


**Strength And Weaknesses:**

  - Strength
    - The use of EBM is novel.
    - The segmentation performance is better than slot-attention
  - Weakness
    - The paper mainly focuses on segmentation performance leaving the reconstruction quantity untouched.
    - While the generative object-centric learning model can sample objects with various appearances, EBM does not support the generation of any form (at least not trivial).
    - While the beta-VAE-based model encourages feature disentanglement, EBM does not explicitly encourage such behavior.
    - And the above weakness are not discussed.
    - It is not entirely clear to me the meaning of "need for specifically-tailored neural network architecture or excessive generative modeling assumption".
    Could the author explain, for example, how IODINE is specifically-tailored or has excessive generative modeling assumptions?

**Summary Of The Paper:**

The paper introduces an energy-based model object-centric learning pipeline.
Energy functions are designed to be permutation invariant.
Experiment results show that scenes are decomposed reasonably well.
Energy function algebra leads to meaningful scene editing results.

**Summary Of The Review:**

  - While the EBM model is not as versatile as the generative object-centric learning model, I agree that it could be a strong object-centric learning candidate.
  - However, I hope the author can explain more about their assertion about their main advantages over previous works.

---

> ### Author Response · Authors · 2022-11-19
> **Response to reviewer tTco**
>
> We appreciate the reviewer's useful feedback on our work, and further thank the reviewer for finding our work novel, clear-written, and could be a strong object-centric learning candidate. Below, we address the reviewer's concerns in turn:
>
> > **Q1** The paper mainly focuses on segmentation performance leaving the reconstruction quantity untouched.
>
> We agree that more evaluation metrics can be helpful. We have added more evaluation results of the mean reconstruction error (MSE) and mean segmentation covering (mSC, [2]) metrics in the appendix.
>
> > **Q2** While the generative object-centric learning model can sample objects with various appearances, EBM does not support the generation of any form (at least not trivial).
>
> Since our primary goal is to learn object-centric representations, we focus on framing our proposed EGO model as a stand-alone object-centric learning module that can be easily plugged into any task-specific models. To this end, we did not attempt to build a full generative model that can do free-form unconditional scene generation. Instead, we show that we can leverage the learned energy functions and recompose them to perform conditional scene generation and manipulation, which is non-trivial to achieve with other baseline approaches.
>
> > **Q3** While the beta-VAE-based model encourages feature disentanglement, EBM does not explicitly encourage such behavior.
>
> We agree that unlike the IODINE model which takes disentanglement into account, our model does not explicitly encourage the learning of disentangled factors of variations. The main motivation of our work is to explore object-centric learning with minimal parametric assumptions, which is drastically different from the line of VAE-based models (e.g. IODINE, MONet) where explicit assumptions of the data generation process are made, such as a disentangled latent space.
>
> > **Q4** It is not entirely clear to me the meaning of "need for specifically-tailored neural network architecture or excessive generative modeling assumption".
>
> By "specifically-tailored neural network architecture", we mean methods like slot-attention which invents a novel inverted attention mechanism to explicitly introduce competition among slots. By "excessive generative modeling assumption", we mean methods like IODINE which explicitly assumes parametric forms about the underlying data generation process. We take a different principle in this work, to learn an energy-based model which is less restrictive in functional form and can avoid the need for tailor-made architectures or inference update rules.

---

> > ### Comment · Reviewer_tTco · 2022-11-30
> > **Thanks**
> >
> > The new results show that the proposed method is on par with previous literature, which demonstrates the potential.
> >
> > I am not sure about the significance of "avoid the need for tailor-made architectures or inference update rules".
> > Please note that a specific form of parametrization does not necessarily mean limited expressiveness or capacity.
> > The proposed model itself may also benefit from carefully designed architecture in the future.
> >
> > As a matter of fact, the MCMC process can be quite time-consuming making scaling up network architecture more difficult.
> >
> > In my mind, a vanilla VAE/AE-based model performs inference in the coarsest fashion.
> > IODINE employs iterative refinement to perform a more fine-grained variational inference leading to better inference results.
> > The same goes for slot attention.
> > The proposed model here is an inference process at its finest level.
> > However, I didn't see concrete advantages in terms of the actual performance.
> >
> > Again, I do very much appreciate the novelty here. I wish I can give a weak acceptance but there is no such option this time.

---

### Official Review · Reviewer_1fbA · 2022-10-23

**Confidence:** 4
**Correctness:** 4
**Technical Novelty And Significance:** 4
**Empirical Novelty And Significance:** 4
**Recommendation:** 8

**Clarity, Quality, Novelty And Reproducibility:**

The paper is clear, of high quality, and novel. The authors provide some implementation details in the appendix. However, I would still like the authors to release the code at some point because some details e.g., how exactly the MCMC sampler should be implemented in code and how the gradient flows to the EGO’s parameters is not obvious to me.

**Strength And Weaknesses:**

### Pros

1. Energy-based modeling is an important framework and the question of whether this approach can be used to learn slots is an interesting one. In this regard, the experiment results are useful — showing that it can perform comparably or somewhat better than the previous methods — and thus has the potential to spark a new line of exploration in the object-centric learning community.
2. The proposed encoder is almost entirely a transformer. I believe this is a good thing given that the transformer is currently a default model in the community and is perhaps more well-understood than a specifically-tailored architecture. This may also make their encoder applicable more generally beyond the image domain unlike some previous models like IODINE.
3. Useful ablations showing the model to be robust to various hyper-parameters.
4. The scene addition and subtraction via summing/subtracting the energy functions is interesting/surprising.
5. Generalizes well to OOD number of objects and unseen object style.
6. Has a probabilistic interpretation and thus the slots may capture the uncertainty.

### Weaknesses/Questions

1. While I believe in the promise of EGO’s probabilistic interpretation (i.e. modeling uncertainty in the partially-observable environments) there is a lack of some experiment that explores this aspect. Authors may consider showing a qualitative experiment of multi-stability in Tetris dataset similar to that shown in the IODINE paper.
2. Some unclear aspects:
    1. In the conclusion section, there is a line saying “minimal assumptions on the generative process…”. This line is a bit unclear to me. Is this referring to the fact that the model does not apply a prior on the latents like IODINE does?
    2. Slot-attention performs an explicit spatial attention and mean-pooling to bind low-level information into object slots. The proposed model does this implicitly via the transformer. What are the implications of this design difference?
    3. How does the gradient flow backward via the MCMC sampling steps and to the EGO’s parameters.
3. Another unexplored question is whether the encoder can be adopted to deal with complex scenes in the same way as slot attention could be adopted in SLATE or SAVi.

### Minor Comments/Questions

1. I am wondering why IODINE is not shown in the property prediction experiment and also why GENESIS is omitted from the ARI result in Table 1.
2. The bar colors of various models can be made consistent across Fig. 2 and Fig 5(a).
3. The word ‘controllable’ in the title could be confusing to some because it might imply steerable/conditional slots such as that learned by SAVi. As I understand, ‘controllable’ seems to be about controlling scene generation and not controlling the slot representation itself.
4. In the section on ‘Increasing the number of objects’, it seems that the referred figure should be 4(c) and not 4(b). Also, in Fig. 4(c), the Y-axis range may be set to 0-100 to better highlight the fact that the performance deterioration is not large when testing on more objects.

**Summary Of The Paper:**

The paper proposes a novel encoder that, given an image, learns object-centric slot representations using the EBM framework. The encoder computes an image-conditioned energy function $E(\mathbf{z}_1, \ldots, \mathbf{z}_K, \mathbf{x}; \boldsymbol{\theta})$ that assigns an energy value to slot representations $\mathbf{z}_1, \ldots, \mathbf{z}_K$. It then applies gradient-based MCMC sampling on the energy function to obtain the slots. Crucially, the proposed energy function $E(\mathbf{z}_1, \ldots, \mathbf{z}_K, \mathbf{x}; \boldsymbol{\theta})$ is invariant to the ordering of the slots $\mathbf{z}_1, \ldots, \mathbf{z}_K$. This order-invariance is achieved via two approaches: 1) by summing a per-slot energy function, OR, 2) by adopting a transformer that treats the slots as a set. The learning signal for training the model parameters comes from a reconstruction objective where a pixel-mixture decoder reconstructs the image by decoding the learned slots.

**Summary Of The Review:**

I think the paper is novel and interesting and should be shared with the community. I support the acceptance of the paper.

---

> ### Author Response · Authors · 2022-11-19
> **Response to reviewer 1fbA (Part 2)**
>
>
> > **Q7** The bar colors of various models can be made consistent across Fig. 2 and Fig 5(a).
>
> Thanks for your suggestion. We have updated the manuscript to make the bar colors consistent across the two figures.
>
> > **Q8** The word ‘controllable’ in the title could be confusing to some because it might imply steerable/conditional slots such as that learned by SAVi. As I understand, ‘controllable’ seems to be about controlling scene generation and not controlling the slot representation itself.
>
> We can confirm that the reviewer's understanding is correct. We have added a few words to clarify the context of the word "controllable" in section 4.2.
>
> > **Q9** In the section on ‘Increasing the number of objects’, it seems that the referred figure should be 4(c) and not 4(b). Also, in Fig. 4(c), the Y-axis range may be set to 0-100 to better highlight the fact that the performance deterioration is not large when testing on more objects.
>
> 1. Thank you for pointing out the typo. We have fixed the typo in the revised version.
> 2. Thanks for the suggestion on the Y-axis range. We tried to set the range to 0-100 but found that the difference between different groups of bars becomes visually much less significant. Thus we decided to keep the original limited Y-axis range.
>
> > **Q10** The paper is clear, of high quality, and novel. The authors provide some implementation details in the appendix. However, I would still like the authors to release the code at some point because some details e.g., how exactly the MCMC sampler should be implemented in code and how the gradient flows to the EGO’s parameters is not obvious to me.
>
> We provided our code in the supplemental material, which includes the implementation of everything needed to reproduce the results in this work. We will also release the code on Github in the future upon publication.

---

> ### Author Response · Authors · 2022-11-19
> **Response to reviewer 1fbA (part 1)**
>
> Thank you for your supportive review and detailed feedback. Moreover, thank you for bringing up several points for improving the manuscript which we found very helpful. We appreciate that you found our work clear, high-quality, and novel. We now address each of the points in your review in turn, and have adjusted the manuscript with the corresponding clarifications:
>
> > **Q1** While I believe in the promise of EGO’s probabilistic interpretation (i.e. modeling uncertainty in the partially-observable environments) there is a lack of some experiment that explores this aspect. Authors may consider showing a qualitative experiment of multi-stability in Tetris dataset similar to that shown in the IODINE paper.
>
> Thank you for your valuable suggestion. We agree that the multi-modal analysis would be interesting, and we follow the IODINE paper to conduct the multi-modal and multi-stability analysis on the Tetrominoes dataset. We show the qualitative results in Figure.11 in the appendix, where we use an ambiguous scene input to show that our model can generate multiple plausible segmentations. We can see from the drawed samples that the model can learn a multi-modal posterior by modeling the uncertainty with the stochastic sampler.
>
> > **Q2** In the conclusion section, there is a line saying “minimal assumptions on the generative process…”. This line is a bit unclear to me. Is this referring to the fact that the model does not apply a prior on the latents like IODINE does?
>
> By "minimal assumptions", we are trying to highlight the motivation behind our work, which is to avoid the need to make strong assumptions about the functional form of data generating process, which is usually the case in VAE-based models like IODINE. Also, we indeed do not assume a prior on the latent variables. We have made the sentence clearer in the revised version to avoid potential confusion.
>
> > **Q3** Slot-attention performs an explicit spatial attention and mean-pooling to bind low-level information into object slots. The proposed model does this implicitly via the transformer. What are the implications of this design difference?
>
> We show that EGO does not need the ad-hoc modifications to the attention mechanism, as done in slot-attention to normalize the attention weights along the slots dimension, and with vanilla cross-attention in the energy function and gradient-based MCMC sampling methods, we can coordinate the updates among the slots implicitly via the gradient coming from the energy function. In this way, we can avoid the need of extra ad-hoc model design efforts on either the attention mechanism or the update rule for slot representations.
>
> > **Q4** How does the gradient flow backward via the MCMC sampling steps and to the EGO’s parameters.
>
> Similar to bi-level nested gradient-based optimization, we directly backpropagate the reconstruction loss through the MCMC sampling process to update the parameters of EGO. Given the relatively small number of MCMC sampling steps and low dimensionality of the latent variables, the bi-level optimization is computationally tractable and efficient. We also discussed the running-time statistics of EGO compared to other baseline methods in the appendix.
>
> > **Q5** Another unexplored question is whether the encoder can be adopted to deal with complex scenes in the same way as slot attention could be adopted in SLATE or SAVi.
>
> While we are following the same setting in slot-attention for the consideration of experimental datasets and setup, we agree that it would be worth exploring the performance of EGO on more complex scenes. We intend to leave this as future work since scaling up the model to more complex scenes would require more engineering efforts on the model implementation and training, which are mostly orthogonal to the main contributions of this work.
>
> > **Q6** I am wondering why IODINE is not shown in the property prediction experiment and also why GENESIS is omitted from the ARI result in Table 1.
>
> 1. For the property prediction task, we adopt the standard implementation and follow the same setup in the benchmark paper [1], which did not include IODINE for downstream property prediction (Figure.4 in [1]).
> 2. For the ARI results in Table.1. we follow the same setup (CLEVR6, Multi-dSprites, Tetrominoes) in slot-attention for the unsupervised object discovery task. Due to the fact that GENESIS works on another set of datasets (Multi-dSprites, GQN, and ShapeStacks) and the intention to keep the comparison style consistent with Table.1 in the slot-attention work, we did not include GENESIS in the table.
>
> [1] Generalization and Robustness Implications in Object-Centric Learning

---

> > ### Comment · Reviewer_1fbA · 2022-12-03
> > **Thank You**
> >
> > Thank you for your rebuttal! It addresses my main concern about demonstrating uncertainty modeling. For the final version, showing samples on more input images in Fig. 11 would make this stronger. I maintain my score of 8.

---

### Official Review · Reviewer_DPAM · 2022-10-24

**Confidence:** 5
**Correctness:** 4
**Technical Novelty And Significance:** 3
**Empirical Novelty And Significance:** Not applicable
**Recommendation:** 6

**Clarity, Quality, Novelty And Reproducibility:**

This paper is of high quality in terms of readability, clarity, and completeness. It also exhibits a new formulation of object-centric learning with an energy based model. The implementation details and experiment settings are very clear, showing good reproducibility from my understanding.


**Strength And Weaknesses:**

Strength:

(1) The energy-based modelling of object-centric learning minimizes assumptions made on the generative process, relieving neural networks from complicated designs to model visual scenes.

(2) The compositionality nature of the energy function makes it possible to perform scene manipulation with learned object-centric latent variables.

Weakness:
(1) In the Introduction, there is a lack of clear motivation to develop energy-based models compared with existing object-centric learning frameworks such as VAE based (MONet, IODINE, etc) or Attention based (SlotAtt). What are the key advantages of energy-based models over others?

(2) The proposed method somehow takes a similar formulation with SlotAtt, especially EGO-Attention. In experiments, the performance gain is also marginal compared with SlotAtt.

(3) The datasets used for unsupervised object discovery are quite simple. Previous approaches have already achieved very high ARI scores on them. Actually, the ARI metric seems not informative because it’s very easy to get high scores. More general metrics such as AP are suggested, as also pointed out in a very recent work [1].

In addition, evaluation and comparison on more challenging datasets such as ShapeStacks [2], ObjectsRoom [3], or even real-world datasets are very desirable. Otherwise, it’s hard to demonstrate the clear advantages of EGO over other baselines.

[1] Promising or Elusive? Unsupervised Object Segmentation from Real-world Single Images, NeurIPS 2022.
[2] ShapeStacks: Learning Vision-Based Physical Intuition for Generalised Object Stacking, ECCV 2018.
[3] Neural Scene Representation and Rendering, Science 2018.

(4) In the evaluation of generalization capability on unseen object styles, it is claimed that the created OOD data have altered colours, textures and shapes. However, it is not demonstrated qualitatively or quantitatively how and to what extent colour jitter transformation and neural style transfer can alter the colour, texture and shape distributions.

**Summary Of The Paper:**

This paper proposes to learn object-centric representation with an energy-based model. This energy model takes a visual scene and a set of object-centric latent variables as input. Latent variables are inferred from visual observations through gradient-based MCMC sampling, where the gradient is derived from the energy function.

To formulate a permutation-invariant energy function over a set of objects, this paper proposes two variants. The first one is to simply sum over individual energy functions from the set of latent variables. Another is to compose them with an attention mechanism.

The proposed model is firstly evaluated on the unsupervised object discovery task, where it demonstrates good segmentation performance. It also exhibits the ability to perform scene manipulation given the compositionality nature of the energy function. Lastly, the proposed model shows robustness to different hyper-parameters and generalization to unseen data.

**Summary Of The Review:**

This paper formulates a permutation-invariant energy model to effectively learn object-centric representation from visual scenes.  The proposed method is inspiring by modelling object-centric learning from a different perspective. The major concern is its structural similarity with a previous method and its marginal performance gain over baselines. It is strongly suggested to demonstrate its capacity for more challenging scenarios using more informative metrics.

---

> ### Author Response · Authors · 2022-11-19
> **Response to reviewer DPAM**
>
> Thank you for your positive feedback and valuable suggestions. We really appreciate your comments, and we are glad to hear you found that our work is clear-written, novel, and reproducible.
>
> #### **Q1** Key advantages of energy-based models over other approaches in object-centric learning.
> The key motivations behind our energy-based models come from:
> 1. Energy-based models are less restrictive in functional form, compared to other VAE-based models (e.g. MONet, IODINE) where specific assumptions about data generating process must be explicitly made, usually in the form of directed graphical model with tractable likelihood and posterior. The choice of energy-based models offers us much more flexibility in the neural architectures, and we show that with MCMC sampling, we can further avoid the need of additional inference network in VAE-based models which needs extra engineering and algorithmic design efforts.
> 2. Energy-based models in our work show advantages over the slot-attention approach on: i) scene generation, where we show that we can recompose learned energy-functions to generate novel scenes, and ii) model robustness, since we are using stochastic MCMC-based sampler for inference while the slot-attention approach is deterministic, we show that our model is more robust to out-of-distribution samples and can achieve better segmentation performance in the presence of occlusion, compared to the slot-attention approach.
>
> #### **Q2** Comparison with slot attention.
> We highlight the key differences between our work and the slot-attention approach in terms of model design and experimental results below:
> 1. One of the main motivations of our works is to show that with energy-based model formulation, we can leverage existing neural network models (cross-attention in ours) for object-centric learning, while slot-attention is a tailor-made module by inventing an inverted attention mechanism to explicitly introduce competition among slots. The similarity between our work and slot-attention roots from the fact that both are built on top of attention mechanism, with the key difference is that we do not need extra ad-hoc modifications.
> 2. The energy-based models offers additional advantages in scene manipulation and generation, which is non-trivial to achieve with slot-attention approach. We provided more detailed discussion in the response to Q1 above.
>
> #### **Q3** Suggestions on using other evaluation metrics and more challenging datasets.
> 1. Thanks for the additional reference ([1]). Since it went public later than the ICLR submission deadline, we did not have the chance to consider the setup introduced, but we did find it interesting.
> 2. We agree that more evaluation metrics can be helpful. We have added more evaluation results of the mean reconstruction error (MSE) and mean segmentation covering (mSC, [2]) metrics in the appendix. We also additionally conducted experiments on set prediction task, and use AP as the evaluation metric. We show the preliminary experimental results in the table below, where we can see that EGO outperforms slot-attention in terms of the AP metrics.
>
>   | Model | AP$_\infty$ | AP$_1$ | AP$_{0.5}$ | AP$_{0.25}$ | AP$_{0.125}$ |
>   | --- | --- | --- | --- | --- | --- |
>   | Slot MLP | 19.8 | 1.4 | 0.3 | 0.0 | 0.0|
>   | DSPN | 85.2 | 81.1 | 47.4 | 10.8 | 0.6 |
>   | Slot-Attention | 94.3 | 86.7 | 56.0 | 10.8 | 0.9 |
>   | EGO | 95.2 | 88.5 | 58.1 | 11.4 | 1.1 |
>
> 3. For the experimental setup on datasets, we mostly follow the same setting in slot-attention for better comparison against the baseline methods. Our methods can achieve comparable performance on the simpler datasets (CLEVR6 and Tetrominoes) where the ARI scores are nearly perfect. We show that our approach outperforms all baseline methods on the more challenging Multi-dSprites dataset, where more occlusion among objects is present.
> 4. Besides regular evaluation of the segmentation performance, we also conducted extensive evaluation on the robustness and generalization to out-of-distribution scenes. We show that our model is consistently superior to the baseline methods when facing unseen object styles or number of objects, in terms of both segmentation performance and downstream prediction.
>
> [1] Promising or Elusive? Unsupervised Object Segmentation from Real-world Single Images.
>
> [2] GENESIS: GENERATIVE SCENE INFERENCE AND SAMPLING WITH OBJECT-CENTRIC LATENT REPRESENTATIONS
>
> #### **Q4** Demonstrating qualitatively or quantitatively how and to what extent colour jitter transformation and neural style transfer can alter the colour, texture and shape distributions.
> As we adopt the proposed implementation of the color jitter transformation and neural style transfer from [3], we refer the readers to Figure.11 in original paper [3] for the qualitative results of the domain shifts effects. We have also made the pointer clearer in our appendix.

---

### Author Response · Authors · 2022-11-19
**Common response and revision summary**

We thank all reviewers for the positive assessment of our work and helpful suggestions for improvements. We have taken these reviews seriously and have updated our manuscript to address the concerns, and all edits made to the manuscript are highlighted in blue. We also summarize the common responses and revisions below.

1. **Additional experimental evaluation results**. As suggested by reviewer DPAM and tTco, besides the ARI scores presented in the manuscript, we include more evaluation results to examine the reconstruction quality in terms of MSE and the segmentation performance in Figure.10 in the appendix.
2. **Additional experiments on more tasks**. As suggested by reviewer vs6t and DPAM, we conducted additional experiments on set prediction, to investigate the performance of EGO on tasks beyond object discovery, as well as the flexibility in integration into other models beyond auto-encoder.
3. **Additional qualitative analysis of the multi-modal posterior**. As suggested by reviewer vs6t and 1fbA, we include an additional qualitative analysis of the multi-modal posterior in Figure.11 in the appendix, following a similar setup as done in IODINE.
4. **Improving clarity on the description of previous works and our motivations**. As pointed out by reviewer vs6t and tTco, we have revised the introduction to make the description of previous works more clear and more accurate. In the responses to the specific comments, we also provided a more detailed discussion of the motivations of our work and the trade-offs made by the proposed formulation.

---

### Decision · Program_Chairs · 2023-01-20

**Decision:**

Accept: poster

**Justification For Why Not Higher Score:**

The method does not perform clearly better than previous alternatives.

**Justification For Why Not Lower Score:**

The proposed formulation is very novel and will help clarify what matters and what not for slot computation.

**Metareview: Summary, Strengths And Weaknesses:**

This paper proposes to learn object-centric representation with an energy-based model. This energy model takes a visual scene and a set of object-centric latent variables as input. Latent variables are inferred from visual observations through gradient-based MCMC sampling, where the gradient is derived from a permutation invariant energy function.
The proposed model is evaluated on the unsupervised object discovery task, where it demonstrates good segmentation performance, comparable to previous models of slot attention. It also exhibits the ability to perform scene manipulation given the compositionality nature of the energy function.
All reviewers agreed on the novelty of the proposed formulation.
They raised concerns regarding i) the advantages of the energy-based framework over previous methods, such as, slot attention, ii) visualization of alternative samples from the energy-based model iii) additional evaluation metrics to be reported, iv) application of the model in more complex setups, and potentially in the real world, v) discussion of the computational challenges of the proposed framework.

The rebuttal submitted by the authors addressed many of these concerns. The authors are encouraged to add all the additional clarifications  and experiments to the paper. Given the novelty of the work, and its comparable performance with alternatives, we believe this is a valuable read for the ICLR audience.


**Note From Pc:**

if the above contains the word "oral" or "spotlight" please see: "oral" presentation means -> notable-top-5% and "spotlight" means -> notable-top-25%. As stated in our emails, we are disassociating presentation type from AC recommendations

**Summary Of Ac-Reviewer Meeting:**

N/A